# Glycogen branching enzyme controls cellular iron homeostasis via Iron Regulatory Protein 1 and mitoNEET

Nhan Huynh[1], Qiuxiang Ou[1], Pendleton Cox[1], Roland Lill [2,3] & Kirst King-Jones [1*]

Iron Regulatory Protein 1 (IRP1) is a bifunctional cytosolic iron sensor. When iron levels are normal, IRP1 harbours an iron-sulphur cluster (holo-IRP1), an enzyme with aconitase activity. When iron levels fall, IRP1 loses the cluster (apo-IRP1) and binds to iron-responsive elements (IREs) in messenger RNAs (mRNAs) encoding proteins involved in cellular iron uptake, distribution, and storage. Here we show that mutations in the *Drosophila* 1,4-Alpha-Glucan Branching Enzyme (*AGBE*) gene cause porphyria. *AGBE* was hitherto only linked to glycogen metabolism and a fatal human disorder known as glycogen storage disease type IV. AGBE binds specifically to holo-IRP1 and to mitoNEET, a protein capable of repairing IRP1 iron-sulphur clusters. This interaction ensures nuclear translocation of holo-IRP1 and down-regulation of iron-dependent processes, demonstrating that holo-IRP1 functions not just as an aconitase, but throttles target gene expression in anticipation of declining iron requirements.

[1] Department of Biological Sciences, University of Alberta, G-504 Biological Sciences Bldg, Edmonton, Alberta T6G 2E9, Canada. [2] Institut für Zytobiologie und Zytopathologie, Philipps-Universität Marburg, Robert-Koch-Strasse 6, 35032 Marburg, Germany. [3] LOEWE Zentrum für Synthetische Mikrobiologie SynMikro, Philipps-Universität Marburg, Hans-Meerwein-Straße, 35043 Marburg, Germany. *email: kirst.king-jones@ualberta.ca

Iron is an essential trace element for nearly all organisms and cells, because iron co-factors, most commonly in the form of haem and iron–sulfur (Fe–S) clusters[1,2], are required for a wide range of protein functions. While essential, free iron is also highly reactive and cytotoxic, and thus its acquisition requires tight regulation by cells[2]. IRP1 is a bifunctional protein because it can reversibly bind to Fe–S clusters[3]. Under iron-replete conditions, the protein forms holo-IRP1 and acts as a cytosolic aconitase, an enzyme that interconverts citrate and isocitrate. When cellular iron levels drop, IRP1 loses its Fe–S cluster and assumes a different conformation[4], apo-IRP1, which then binds to and regulates specific target mRNAs containing iron-responsive elements (IREs). This action can either block translation or enhance transcript stability, depending on the location of the IRE, with a net outcome that promotes increased cellular iron availability and trafficking[5].

Current models for studying cellular iron homeostasis are limited in the sense that they have either static, or at best, linearly increasing iron requirements (cell cultures and developing erythrocytes, respectively)[2,6]. We introduce here the *Drosophila* prothoracic gland (PG) as a model to study highly dynamic iron requirements. The PG is an endocrine tissue mainly devoted to the production of steroid hormones in developing insects. In both vertebrates and insects, the synthesis of steroid hormones is largely dependent on enzymes that require haem and Fe–S clusters[7,8]. In *Drosophila*, the PG is the principal steroid-producing gland, and part of a larger endocrine tissue called the ring gland. During larval development, the PG produces pulses of the steroid hormone ecdysone (E), which acts as a recurring systemic signal that controls gene expression in target tissues to coordinate hatching, moulting, and metamorphosis (Supplementary Fig. 1). The first step of E biosynthesis is carried out by Neverland, which harbours an Fe–S cluster[9], while all but one of the following reactions are carried out by cytochrome P450 proteins, which require haem as an obligate co-factor (Fig. 1a). The last larval stage of *Drosophila* development is accompanied by exceedingly high expression of ecdysone-producing enzyme ("Halloween") genes (Supplementary Fig. 1)[8], indicating that the PG requires substantial amounts of iron, which can be visualized by staining for ferric iron (Fig. 1b). PG cells have fluctuating iron and haem demands because they must match the rise and fall of Halloween enzyme levels with appropriate production rates of iron co-factors. Thus, the PG represents a powerful model to study mechanisms by which cells control dynamic changes in cellular iron and haem requirements. We show here that the *Drosophila* Glycogen Branching Enzyme (AGBE) is a regulator of iron homeostasis. AGBE interacts physically with the holoform of Iron Regulatory Protein 1A (IRP1A) and Cisd2, an ortholog of vertebrate mitoNEET. This synergistic interaction ensures that holo-IRP1A remains functional. Further, we show that holo-IRP1A has a surprising new role in the nucleus, where it transcriptionally downregulates genes acting in steroid hormone biosynthesis as well as iron and heme metabolism.

## Results

**Loss of glycogen branching enzyme causes porphyria-like phenotypes**. The haem biosynthetic pathway is highly conserved in metazoans and fungi and comprises eight enzymatic steps that convert glycine and Succinyl-CoA to mature haem (Fig. 1c)[10]. Exposure to air and UV light isomerizes porphyrinogen rings, first produced in step 4, into autofluorescing porphyrins, but incorporation of iron into protoporphyrin IX results in non-fluorescing haem (Fig. 1d)[10,11]. We noticed the presence of red autofluorescence in the PG when we exposed larvae from four RNAi lines to UV light (targeting *Updo*, *Ppox*, *spz5*, and *AGBE*),

all of which had been identified in two unrelated PG-specific RNA interference (RNAi) screens[7,8]. Also, the ring glands were enlarged compared to time-matched controls and had a red-brown appearance under brightfield light (Supplementary Fig. 2). A fifth RNAi line, *Nos^{IR-X}*, targeting the *nitric oxide synthase* (*Nos*) gene, had been reported to produce large red-brown PGs[12], and when re-examined by us, also showed red autofluorescence (Fig. 1e). Consistent with their role in haem biosynthesis, depleting Updo (=vertebrate UROD, Supplementary Fig. 3) and Ppox caused protoporphyrin accumulation in the PG. This is equivalent to what occurs in patients afflicted with porphyria, a group of rare diseases impairing haem biosynthesis[13]. PG-specific *Alas*-RNAi, on the other hand, disrupts haem synthesis prior to porphyrinogen ring formation, and therefore lacked the auto-fluorescence, but causes enlarged ring glands (Fig. 1e). We then sought to validate the three remaining lines, *Nos^{IR-X}*, *AGBE^{IR1}*, and *spz5^{IR}*, since their relationship to haem biosynthesis was intriguing. We were unsuccessful in finding independent evidence for the *Nos^{IR-X}* and *spz5^{IR}* lines, suggesting that the phenotypes were caused by off-target effects. However, a second, non-overlapping RNAi line targeting *AGBE*, *AGBE^{IR2}*, caused similar phenotypes (Fig. 1f). *AGBE* encodes a glycogen branching enzyme, which is an essential enzyme that acts in glycogen biosynthesis[14]. There are, however, no reports that link glycogen branching enzymes to iron or haem homeostasis. Therefore, we further validated these results by using CRISPR/CAS9 to replace the endogenous *AGBE* gene with a genomic copy that was flanked by Flippase (FLP) Recombinase Target (FRT) sites, and where the last exon extended into a Flag- or Myc-tag (*AGBE^{FCF}* and *AGBE^{FCM}*, Supplementary Fig. 4). Excision of the conditional *AGBE^{FCF}* allele via PG-specific expression of FLP confirmed the *AGBE*-RNAi phenotypes, as we observed strong autofluorescence with overall higher penetrance than the two RNAi lines, since no adults eclosed compared to 4.6% in homozygous *PG > AGBE^{IR1}* animals (Fig. 1f, g).

We reasoned that a lack of cellular or mitochondrial iron could disrupt haem production and may explain the porphyria phenotype in *AGBE*-loss-of-function lines. Remarkably, upon rearing *PG > AGBE^{IR1}* and *PG > FLP; AGBE^{FCF}* larvae on an iron-supplemented diet, the autofluorescence was absent (Fig. 1f), and we observed that ~40–50% of the larvae now developed into phenotypically normal adults (Fig. 1g). In agreement with this, adding the iron chelator bathophenanthroline sulfate (BPS) slightly decreased survival rates, while adding both iron and BPS to the diet reversed the rescue seen by iron alone, confirming that BPS is an effective tool to reduce available iron. *AGBE*-RNAi larvae were uniquely rescued by dietary iron, since neither *Alas*-, *Updo*-, *Ppox*-, *Nos*- or *spz5*-RNAi lines benefited from iron supplementation (not shown). *AGBE* expression was moderately upregulated under iron-chelating conditions, consistent with the idea that the gene partakes in cellular iron homeostasis (Fig. 1h). Next, we tested whether disrupting four other glycogen biosynthesis genes via PG-specific RNAi would phenocopy AGBE-depletion (Supplementary Fig. 5). This neither caused autofluorescence nor significant lethality. However, ubiquitous expression of RNAi targeting these glycogen biosynthesis genes caused widespread larval lethality, confirming that all RNAi transgenes in these lines were expressed, which suggested that the disruption of glycogen biosynthesis per se in the PG did not cause any iron- or haem-related phenotypes, but was a unique feature of AGBE.

**Glycogen branching enzyme physically interacts with Iron Regulatory Protein 1 (IRP1)**. AGBE is the single orthologue of vertebrate GBE1 (Glycogen Branching Enzyme 1), and the two

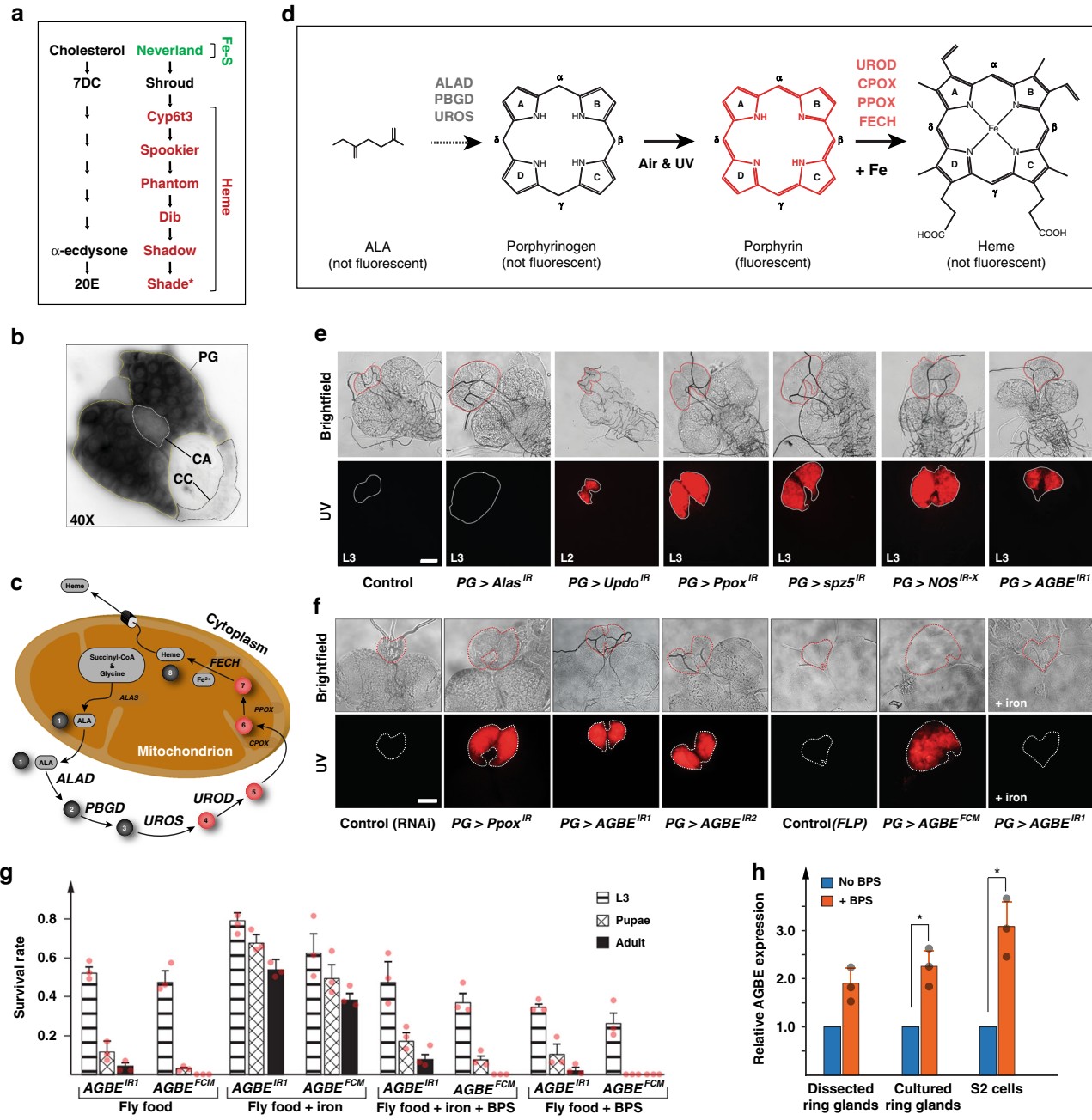

**Fig. 1** Disruption of haem biosynthesis in the *Drosophila* prothoracic gland (PG). **a** Ecdysone biosynthetic pathway converts cholesterol to α-ecdysone, which is metabolized to 20OH-ecdysone in target cells by Shade (*not or lowly expressed in the PG). All enzymes except for Shroud require iron co-factors in the form of iron–sulfur (Fe–S) clusters or haem. **b** Stain for ferric (non-haem-bound) iron in the ring gland. The corpus allatum (CA) and the corpora cardiaca (CC) are neighbouring glands fused to the PG. **c** Haem biosynthesis pathway in metazoans and yeast. Red circles represent protoporphyrin intermediates that autofluoresce. **d** Autofluorescence of porphyrins occurs through isomerization of porphyrinogens exposed to air and UV light. **e** UV exposure of dissected ring glands from RNAi lines (designated as gene[IR]) from second (L2) or third (L3) instar stages. *Alas*, *Updo*, and *Ppox* encode haem-synthesizing enzymes. spz5: spaetzle5, Nos: nitric oxide synthase, AGBE: 1,4-Alpha-Glucan Branching Enzyme. Scale bar = 250 μm. **f** UV exposure of dissected ring glands isolated at 40 h after the L2/L3 moult (~8 h prior to pupariation in controls). RNAi lines *AGBE[IR1]* and *AGBE[IR2]* target distinct regions of the *AGBE* mRNA. *AGBE[FCM]* is a conditional CRISPR-knock-in allele that can be excised in a tissue-specific manner via the expression of Flippase (FLP) recombinase (Supplementary Fig. 4). + iron: larvae were reared on a diet containing ferric ammonium citrate (FAC) as an iron supplement. Scale bar = 250 μm. **g** Survival of *AGBE[IR1]* and *AGBE[FCM]* larvae fly food supplemented with iron (FAC) or an iron chelator, bathophenanthroline sulfate (BPS). Error bars represent standard deviation. Three biological replicates, with each sample containing 50 individuals. **h** Relative *AGBE* mRNA expression levels. Dissected ring glands: isolated from L3 reared on media ± BPS. Cultured ring glands: isolated from L3 reared on normal media, but then transferred to buffer containing ± BPS. S2 cells: Schneider 2 cells grown on medium ± BPS. mRNA levels were analysed via quantitative real-time PCR. For primers see Table 3. Asterisk indicates a *P*-value < 0.05 based on the Student's *t* test. Error bars represent 95% confidence intervals. Each of the three biological replicates was tested three times. Source data are provided as a Source Data file.

proteins are 61% identical. A search of protein–protein interaction databases[15,16] revealed that human GBE1 physically interacts with IRP1. Vertebrates have two *IRP* genes, IRP1 and IRP2, but only IRP1 can switch between the aconitase and the RNA-binding form, while IRP2 lacks the Fe–S cluster and is constitutively RNA-binding[3]. *Drosophila* does not have the *IRP2* gene but harbours two *IRP1* genes (IRP1A and IRP1B) (Supplementary Fig. 6). Only IRP1A has been shown to switch from holo- to the IRE-binding apoform, while IRP1B is believed to act only as an aconitase, as it failed to bind canonical IREs[17]. The reported interaction between GBE1 and IRP1 raised the possibility that *Drosophila* AGBE and human GBE1 function in the regulation of iron homeostasis by modulating IRP1 activity. Using a cell culture approach, we established that the interaction also occurred in *Drosophila*, namely between AGBE and IRP1A (Fig. 2a), and we recapitulated the interaction between human IRP1 and GBE1 in the same system (Fig. 2b). AGBE interacted robustly with wild type IRP1A, as well as with IRP1A$^{3R3Q}$, which carries three point mutations (R549Q, R554Q, R793Q, see Supplementary Fig. 4) predicted to disrupt RNA-binding[18]. Strikingly, a single point mutation that prevents Fe–S cluster binding to IRP1A (predicted to generate constitutively RNA-binding IRP1A$^{C450S}$) abolished the interaction with AGBE (Fig. 2a), suggesting that holo-IRP1A is the in vivo binding partner of AGBE. This was paradoxical, as *AGBE* mutations caused iron-deficiency phenotypes, but holo-IRP1A has no known roles in iron homeostasis as it is believed to only act as an aconitase. To examine whether the IRP1A$^{3R3Q}$ and IRP1A$^{C450S}$ forms acted as predicted, we tested their in vivo ability to bind an IRE-containing mRNA (*SdhB*) and whether either of them had aconitase activity (Fig. 2c, d). As expected, the IRP1A$^{3R3Q}$ form displayed strongly reduced mRNA-binding compared to both the wild type and the IRP1A$^{C450S}$ forms (Fig. 2c). Likewise, both knocked-in and transgenic alleles of IRP1A$^{C450S}$ resulted in significantly reduced aconitase activity compared to wild type IRP1A and the IRP1A$^{3R3Q}$ form (Fig. 2d and Supplementary Fig. 7).

To further substantiate the interaction between AGBE and holo-IRP1A, we carried out a series of MALDI-TOF-based mass spectrometry (MS) experiments (see later section). We also sought to validate this interaction by genetic means. For the latter, we tested whether animals with PG-specific loss of *AGBE* function could be rescued by expressing transgenic wild type *IRP1A* or *IRP1A$^{C450S}$* (Supplementary Table 1). Remarkably, wild type *IRP1A* rescued both the larval lethality (Fig. 2e, f) as well as the porphyria phenotype of animals that lacked functional AGBE (Fig. 2g), while *IRP1A$^{C450S}$* was completely ineffective (Fig. 2e, g). It was possible that the *IRP1A$^{C450S}$* allele was not functional, despite differing only in a single point mutation from *IRP1A*. However, when we expressed *IRP1A$^{C450S}$* in other genetic backgrounds, we observed dramatic rescue of *PG > NOS$^{IR-X}$* RNAi animals with respect to both the lethality (Fig. 2f) and protoporphyrin accumulation (not shown). Also, our immunoprecipitation results showed that IRP1A$^{C450S}$ is RNA-binding (Fig. 2c). Taken together, these data demonstrated that IRP1A$^{C450S}$ was fully active but not sufficient to compensate for the iron deficiency in AGBE-depleted animals, suggesting that holo-IRP1A has functions beyond the aconitase that are important for iron homeostasis. To test whether the aconitase function of holo-IRP1A had unexpected essential functions, we attempted rescuing *AGBE$^{FCM}$* mutants with the aconitase-only form of IRP1A (IRP1A$^{3R3Q}$), as well as a cytosolic and mitochondrial version of yeast aconitase (*YAco1$^{WT}$* and *YAco1$^{ΔSp}$*, respectively), neither of which can switch to the RNA-binding form[18,19]. None of these approaches rescued the loss-of-AGBE-function phenotypes (Fig. 2g), indicating that both holo- and apo-IRP1A were required for survival. Lastly, we

crossed human *IRP1* (*hIRP1*) and *IRP2* (*hIRP2*) into the *AGBE$^{IR1}$*-RNAi strain. Consistent with the above findings, only hIRP1 (equivalent to IRP1A, Supplementary Fig. 6) could fully rescue *AGBE$^{IR1}$* larvae, while constitutively RNA-binding hIRP2, was much less effective (Fig. 2g), albeit more efficient than IRP1A$^{C450S}$, suggesting partial rescue by hIRP2.

Since no no null mutations were available for *IRP1A* or *IRP1B*, we needed to establish that a) these genes had indeed roles in *Drosophila* iron regulation and if so, b) whether *IRP1A* and *IRP1B* had distinct roles in controlling cellular iron levels, c) whether these genes were required in the PG, and d) whether this would phenocopy AGBE-depletion in the PG. We first disrupted both *IRP1A* and *IRP1B* in the *PG* via RNAi (*PG > IRP1A$^{IR}$* and *PG > IRP1B$^{IR}$*). On regular fly food, neither RNAi line resulted in obvious phenotypes. However, when flies were reared on iron-depleted fly food for three generations, *PG > IRP1A$^{IR}$* animals displayed significant larval lethality, with a concomitant appearance of red autofluorescence in the larval PG (Fig. 3a). Control and *PG > IRP1B$^{IR}$* populations did not exhibit lethality until the 5th generation, and larvae never showed any autofluorescence (not shown and Supplementary Fig. 8). To confirm these data, we used two approaches. First, we generated a Flag-tagged and FRT-flanked knock-in allele of endogenous *IRP1A* (*IRP1A$^{FCF}$*), allowing us to excise the gene via PG-specific expression of *FLP* (Supplementary Fig. 4). This approach resulted in red-fluorescing PGs in *PG > FLP;IRP1A$^{FCF}$* larvae that were switched from iron-replete to BPS-containing media (Fig. 3b). However, homozygous *IRP1A$^{FCF}$* flies were not viable on regular fly media, indicating that the inserted *FRT* sites had generated a loss-of-*IRP1A*-function allele. Therefore, we employed a second CRISPR strategy, where we crossed flies that specifically expressed *CAS9* in the PG[20] to flies expressing two gRNAs that targeted *IRP1A* (Supplementary Table 1). The resulting F1 progeny also displayed PG-specific autofluorescence and 100% lethality on regular fly media (Fig. 3b). To complement these PG-specific lesions with classic mutant analysis, we examined an existing mutant *IRP1A* line (Bloomington #30181). However, we did not consider this allele further as it turned out to be a weak hypomorph. We, therefore, generated deletion mutants for *IRP1A* and *IRP1B* using CRISPR/CAS9, designated here as *IRP1A$^{KO}$* and *IRP1B$^{KO}$*. On regular fly food, *IRP1A$^{KO}$* mutants died as first (L1) and second instar larvae (L2), but were able to develop into phenotypically normal adults when reared on an iron-supplemented diet (Fig. 3c), indicating that IRP1A was essential for responding to iron-poor conditions. In contrast, *IRP1B* mutants revealed no obvious phenotypes under any of the tested conditions. Taken together, these results showed that IRP1A is the principal regulator of cellular iron homeostasis in *Drosophila* and that IRP1A depletion phenocopied the iron-dependent porphyria seen in *AGBE* mutants.

**Subcellular localization of apo- and holo-IRP1.** We then addressed whether holo-IRP1A had unanticipated roles in the regulation of cellular iron homeostasis, since only the holoform interacted with AGBE, and was required to rescue *AGBE* mutants. When we carried out immunolocalisation of *PG > 3xFlag-IRP1A$^{WT}$* and *PG > 3xFlag-IRP1B$^{WT}$* transgenic lines (Supplementary Table 1), we found that both IRP1A and IRP1B were enriched in PG nuclei (Fig. 3d). In stark contrast, expressing the single-point mutation variants *IRP1A$^{C450S}$* or *IRP1B$^{C447S}$* (which abolishes Fe–S-binding in IRP1B, Supplementary Table 1) resulted in predominantly cytoplasmic accumulation of either protein (Fig. 3d). Similarly, we found that human IRP1 localises to PG nuclei as well, while human IRP2 failed to do so (Fig. 3e). This behaviour is consistent with the absence of an Fe–S cluster in

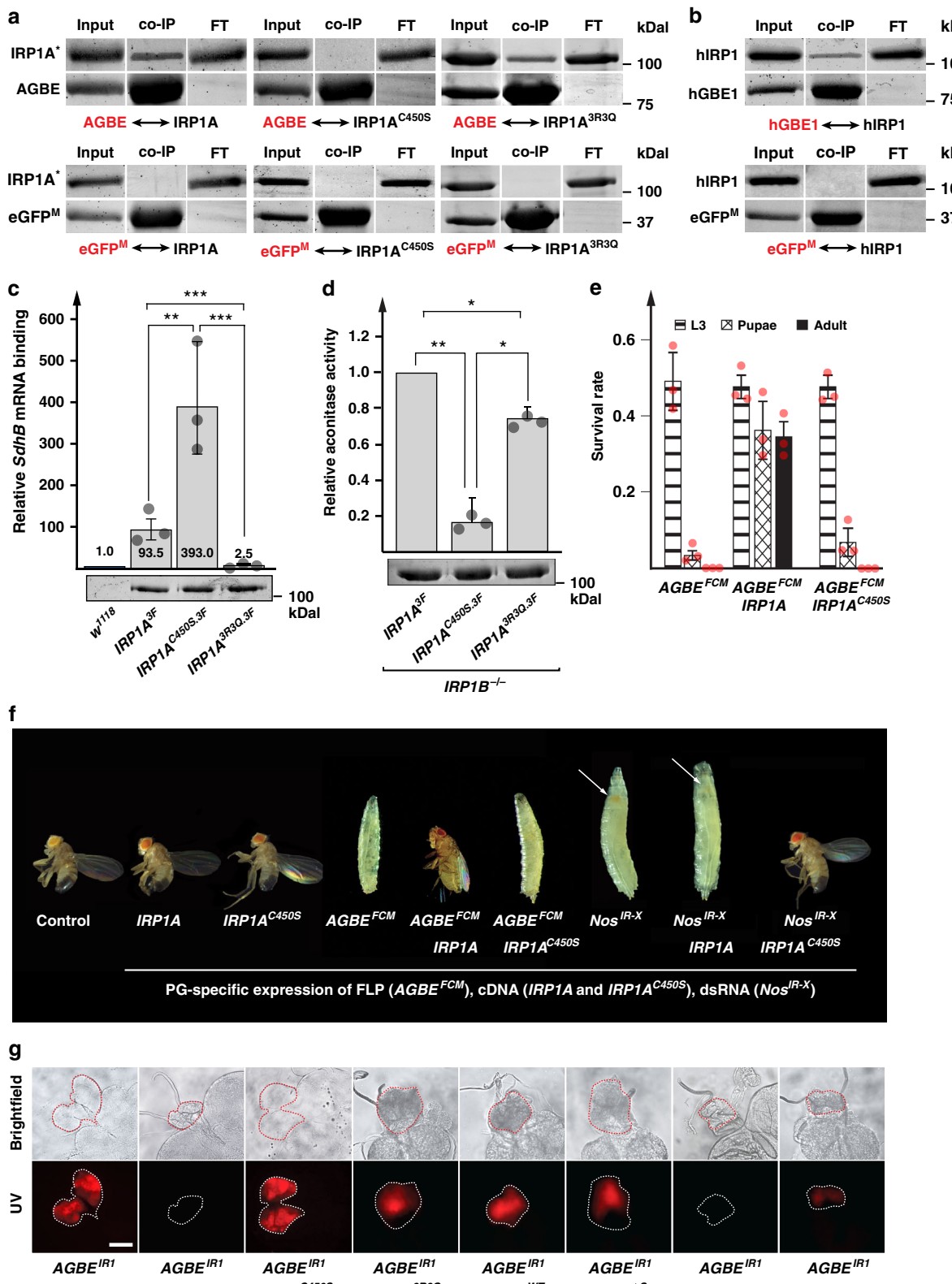

hIRP2, indicating that nuclear accumulation strongly favours holo-IRP proteins.

Given the interaction between AGBE and holo-IRP1A, we wondered whether entry of IRP1A into nuclei was dependent on AGBE. For this, we crossed the Flag-tagged *IRP1A^3F* and *IRP1B^3F* knock-in alleles (Supplementary Fig. 4) into *AGBE* mutants. This approach revealed that IRP1A, but not IRP1B, was dependent on

*AGBE* for nuclear translocation (Fig. 3f), suggesting that IRP1A requires AGBE for maintaining Fe–S clusters, which in turn are needed for nuclear entry. Finally, we determined the subcellular localisation of IRP1A^3F and IRP1B^3F in other tissues. Both IRP1A and IRP1B showed strong cytoplasmic and negligible nuclear presence in the larval salivary gland, while the adjacent fat body displayed predominantly nuclear IRP1A and IRP1B (Fig. 3g).

**Fig. 2** AGBE interacts with IRP1A. **a** Co-transfection of S2 cells with plasmids encoding Flag-tagged IRP1A variants (IRP1A*) and Myc-tagged AGBE followed by immunoprecipitation via anti-Myc antibodies and Western blotting. Names shown in red indicate the protein used as bait. IRP1A: wild type IRP1A, IRP1A$^{C450S}$: constitutively RNA-binding IRP1A, IRP1A$^{3R3Q}$: non-RNA-binding form of IRP1A (Supplementary Table 1). Myc-tagged enhanced GFP (eGFP$^{M}$) served as a negative control. Input lane represents 10% of the sample. Presence of co-immunoprecipitated proteins were tested with anti-Flag antibodies. **b** Like A, but co-transfection of S2 cells with plasmids encoding Flag-tagged human IRP1 (aka Aco1) and Myc-tagged human GBE1, as well as eGFP$^{M}$ as a negative control. **c** Quantitative RNA-immunoprecipitation (RIP). Samples from larvae carrying Flag-tagged knock-in alleles of IRP1A (IRP1A$^{3F}$, IRP1A$^{C450S.3F}$, and IRP1A$^{3R3Q.3F}$) (Supplementary Fig. 4) were normalized via Western blotting to visualize Flag-tagged proteins followed by ImageJ quantification. Western blot of adjusted samples shown below graph. Untagged IRP1A (control line w$^{1118}$) served as a negative control and calibrator (normalized expression = 1). SdhB mRNA harbours a validated IRE[72,73]. Co-immunoprecipitated SdhB mRNA was quantified via qPCR. Error bars represent 95% confidence intervals from three biological replicates. ***$p < 0.001$, **$p < 0.01$, *$p < 0.05$. **d** Aconitase activity. Same IRP1A alleles and normalization procedure as described in **c**, except that IRP1A$^{3F}$ served as the control (normalized to 1). All alleles were crossed into an IRP1B$^{-/-}$ mutant background to eliminate the aconitase activity of IRP1B. Further, we removed mitochondria via ultracentrifugation to reduce the contribution of mitochondrial aconitase. Error bars represent standard deviation from three biological replicates. **$p < 0.01$, *$p < 0.05$. **e** Survival rates of PG > FLP; AGBE$^{FCM}$ animals (Supplementary Fig. 4), which causes Flippase-mediated excision of the AGBE transcription unit specifically in the prothoracic gland (PG). Tested in either the presence or absence of the IRP1A and IRP1A$^{C450S}$ transgenes that are also expressed in a PG-specific manner. Error bars represent standard deviation from three biological replicates (each sample contained 50 individuals). **f** Larval and adult phenotypes of PG > FLP; AGBE$^{FCM}$ and PG > Nos$^{IR-X}$ animals expressing IRP1A$^{C450S}$ or wild type IRP1A transgenes. Arrows point to red-stained PG. **g** Ring glands dissected from PG > AGBE$^{IR1}$ larvae in the presence or absence of the following transgenic cDNAs: IRP1A (wild type IRP1A); IRP1A$^{C450S}$ (constitutively RNA-binding); IRP1A$^{3R3Q}$ (non-RNA-binding); YAco1$^{WT}$: wild type yeast aconitase (mitochondrial); YAco1$^{\Delta Sp}$ (cytoplasmic); hIRP1 & hIRP2: human IRP1 & IRP2. Scale bar = 250 μm. All transgenes are expressed in a PG-specific manner via the Gal4-UAS system. Source data are provided as a Source Data file.

This strongly suggests that nuclear translocation of IRP1 proteins is highly tissue-specific, and not a function of systemic iron load, and hence may reflect tissue-specific iron requirements. Given that human IRP1 also localises to *Drosophila* nuclei, this raises the question of whether vertebrate IRP1 may also enter nuclei in specific tissues or during specific developmental/physiological conditions.

We next sought to identify proteins that would physically interact with IRP1A and IRP1B in order to shed light on the interaction with AGBE and the presence of both IRP1s in nuclei. For this, we immunoprecipitated endogenous or transgenic versions of Flag-tagged AGBE, IRP1A and IRP1B and subjected ring gland and whole-body samples to MS, for a total of 17 conditions (Supplementary Table 2). As controls, we used a total of five wild type samples (which lack Flag-tagged proteins), processed them in parallel to the experimental samples, and removed all proteins found in the controls from the experimental data sets (Supplementary Data 1). Briefly, the interactome for IRP1A indicated extensive interactions with ribosomal proteins and eukaryotic initiation factors, consistent with previous findings[21,22] and IRP's role in regulating translation. We also identified four histone proteins for IRP1A (H4, H2A, H2B, and H2Av) and two for IRP1B (H4 and H2A), consistent with the presence of both IRP1s in nuclei (Fig. 4a and Supplementary Data 1–2). Importantly, endogenously tagged AGBE$^{FCF}$ specifically pulled down a total of 22 proteins from whole-body samples (Fig. 4b), which included IRP1A, IRP1B and Cisd2, an Fe–S protein. Vertebrate Cisd1, Cisd2, and Cisd3 comprise a small family referred to as the NEET proteins[23], which harbour an unusual 2Fe-2S cluster that enables these proteins to transfer their cluster to other proteins[24–26]. *Drosophila* encodes only two NEET proteins, Cisd2 and CG3420 (=> Cisd3), where Cisd2 lies evolutionary between human mitoNEET (encoded by Cisd1) and Naf-1 (encoded by Cisd2)[27], and as such, fly Cisd2 may be functionally related to both.

We also used endogenously tagged IRP1A (IRP1A$^{3F}$) as bait, which co-immunoprecipitated 166 proteins that included AGBE, Cisd2, and IRP1B, as well as two ferritins (iron storage proteins), Fer1HCH and Fer2LCH (Fig. 4a, b and Supplementary Data 1–2). Both AGBE and IRP1A interacted with another glycogen enzyme, glycogen synthase (GlyS), further corroborating that cellular iron homeostasis and glycogen metabolism are physically linked. IRP1B pulled down AGBE and the histones H2A and H4, but not

Cisd2. For IRP1A, all above interactions, with the exception of GlyS, were validated by PG-specific MS (Fig. 4a and Supplementary Data 1). Finally, we further validated these MS data with samples from four fly strains that expressed one of the following transgenes: apo-IRP1A (IRP1A$^{C450S}$); apo-IRP1B (IRP1B$^{C447S}$); non-RNA-binding IRP1A (IRP1A$^{3R3Q}$) and IRP1B (IRP1B$^{3R3Q}$) (Supplementary Tables 1–2). This approach confirmed the results seen with the knock-in alleles, and, importantly, showed that IRP1A$^{C450S}$ failed to interact with AGBE, while Cisd2 interacted with both IRP1A variants, but none of the IRP1B proteins (Supplementary Fig. 9).

Since iron-depletion triggers the switch from holo- to apo-IRP1A, one would predict that this results in cytoplasmic accumulation of IRP1A and should, therefore, alter the interactome of this protein. When we reared flies for three generations on BPS-containing media, we noticed that it takes two generations to purge IRP1A and IRP1B from PG nuclei (Supplementary Fig. 10). In the fat body, however, it takes only one generation for IRP1A to become entirely cytoplasmic, while IRP1B is still nuclear after three generations of iron-depletion (Supplementary Fig. 11), suggesting that IRPs have tissue-specific behaviours. To test whether iron-depletion affected protein-protein interactions of IRP1A, we reared endogenously tagged IRP1A$^{3F}$ flies for two generations on BPS-supplemented food and conducted MS from whole-body larval samples. This strategy reduced the number of co-immunoprecipitated proteins from 166 (no BPS) to 117 (in G1 = one generation BPS) and 30 (in G2 = two generations of BPS) (Supplementary Data 1). Consistent with the cytoplasmic localisation of IRP1A on BPS media, the interaction with histone H2Av and H2B was lost in G1, and none of the four histone proteins were detected in G2, which resembles the pattern seen in the co-immunoprecipitation results with the cytoplasmic IRP1A$^{C450S}$ protein (Supplementary Data 1). Further, binding to AGBE was lost in G1 and G2, consistent with our finding that AGBE only interacts with holo-IRP1A. The interaction between IRP1A and IRP1B was lost in G2, while binding to Fer2LCH was detectable in all conditions. The top-scoring protein in G2 was Cisd2/mitoNEET (Supplementary Data 1), indicating that the IRP1A-Cisd2/mitoNEET interaction was robust even when iron levels were low.

**MitoNEET mutants phenocopy IRP1A and AGBE mutants.** MitoNEET is a homodimeric Fe–S protein that resides in the

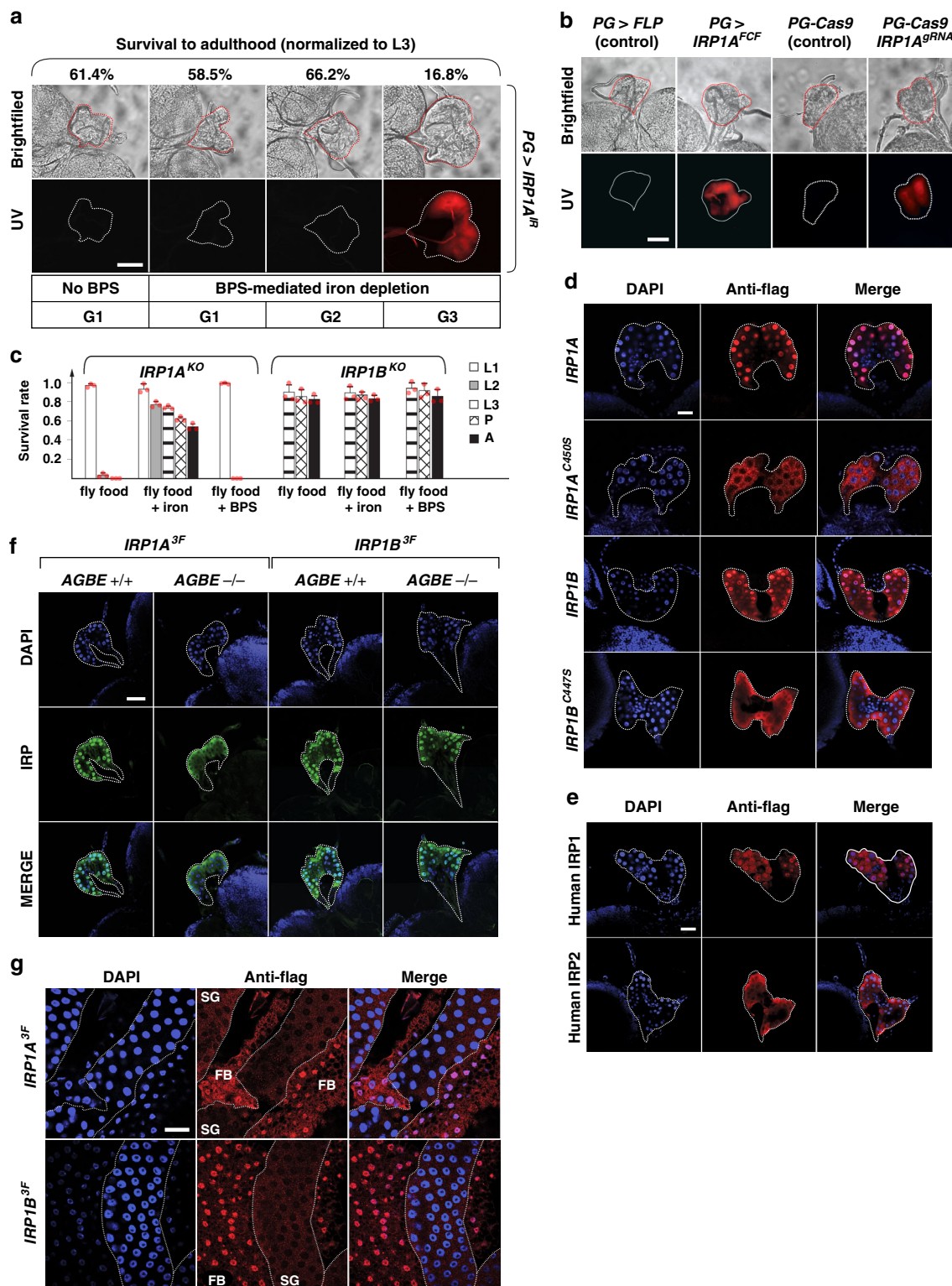

outer mitochondrial membrane, with the Fe–S cluster facing the cytosol[23]. While the exact range of functions for mitoNEET remains unclear, the protein has been shown to act in the repair of oxidatively damaged vertebrate IRP1 Fe–S clusters[23,28]. Since the MS data suggested that Cisd2/mitoNEET represents an important link between AGBE and IRP1A, we sought validation by molecular and genetic means. First, we validated the physical interaction via co-immunoprecipitation assays in *Drosophila*

Schneider 2 cells (S2) cells. This showed that both AGBE and IRP1A physically interact with Cisd2 (Fig. 4c). However, this interaction was ~3-fold enhanced when AGBE and IRP1A were co-transfected together with Cisd2 (Fig. 4c, d), suggesting synergistic interactions between the three proteins.

Next, we examined Cisd2 function by genetic means. When we depleted Cisd2 via RNAi in the PG or examined a *Cisd2* mutant, flies survived on regular food, and displayed no protoporphyrin

**Fig. 3** IRP1 localizes to nuclei. **a** Protoporphyrin accumulation/autofluorescence in prothoracic glands (PG) from *IRP1A*-RNAi (*IRP1A^IR^*) flies reared on iron-depleted (=BPS) media for three generations. Adult survival relative to last larval stage (surviving third instar larvae = L3 = 100%). Scale bar = 250 μm. **b** Protoporphyrin accumulation/autofluorescence in PGs from *IRP1A^FCF^* animals (tissue-specific excision of *IRP1A*, Supplementary Fig. 4) reared on iron-rich medium until L2, after which larvae were switched to BPS-supplemented food. Scale bar = 250 μm. **c** Survival of *IRP1A* and *IRP1B* null mutants (KO = knockout, Supplementary Fig. 4). Error bars represent standard deviation from three biological replicates (each sample contained 50 individuals). **d** Subcellular localization of PG-specific, Flag-tagged IRP1A and IRP1B (*PG > IRP1A/PG > IRP1B* transgenic lines, Supplementary Table 1). DAPI was used to stain DNA/nuclei. Scale bar = 250 μm. **e** Subcellular localization of Flag-tagged transgenic human IRP1 and IRP2 (*PG > hIRP1* and *PG > hIRP2*, Supplementary Table 1) expressed specifically in the PG. Scale bar = 250 μm. **f** Subcellular localization of Flag-tagged proteins encoded by *IRP1A^3F^* and *IRP1B^3F^* knock-in alleles (Supplementary Fig. 4) in control or *AGBE* mutant backgrounds (*AGBE^+/+^ = PG > FLP. AGBE^−/− = PG > FLP; AGBE^FCM^*, Supplementary Fig. 4). Scale bar = 500 μm. **g** Subcellular localization of Flag-tagged proteins encoded by *IRP1A^3F^* and *IRP1B^3F^* knock-in alleles (Supplementary Fig. 4) in the fat body (FB) and salivary gland (SG). Scale bar = 500 μm. Source data are provided as a Source Data file.

accumulation (Fig. 4e, f). On BPS media, however, most (*Cisd2* mutant) or all (*Cisd2*-RNAi) animals arrested development during the third instar and displayed red autofluorescence in the PG (Fig. 4e, f). We then tested whether Cisd2 and IRP1A interacted genetically, and therefore analysed RNAi lines targeting *Drosophila Cisd2*, *IRP1A* and *IRP1B* alone and in combination. None of the individual PG > RNAi larvae displayed any overt phenotypes when reared on regular fly media. However, when we combined *IRP1A*- with *Cisd2*-RNAi, we observed strong synthetic lethality, where none of the larvae reached adulthood, and importantly, all larvae displayed protoporphyrin accumulation in the PG (Fig. 4g), indicating that both proteins participate in the same process. In contrast, the combination of *IRP1B*- with *Cisd2*-RNAi was as ineffective as the individual lines alone. We concluded that the functional importance of the IRP1-mitoNEET interaction is conserved between vertebrates and *Drosophila* and that this process is essential, at least in *Drosophila*. Finally, we tested whether IRP1A and IRP1B could localise to nuclei in a *Cisd2*-mutant background. To test this, we crossed flies that harboured Flag-tagged *IRP1A^3F^* or *IRP1B^3F^* knock-in alleles into a *Cisd2*-mutant background and reared them on fly food in the presence or absence of BPS. On regular fly food, both proteins were nuclear, while exposure to BPS shifted their subcellular distribution to the cytoplasm of the PG (Fig. 4h). Control flies reared for one generation on BPS-containing food still show predominantly nuclear IRP1A and IRP1B in the PG (Supplementary Fig. 10). Taken together, these data indicate that AGBE, Cisd2/mitoNEET, and IRP1A act together to ensure that holo-IRP1A remains functional and can enter the nucleus.

**Nuclear IRP1**. An intriguing possibility is that holo-IRP1 has additional roles in the nucleus that contribute to tissue-specific cellular iron homeostasis. This is supported by the MS data, which indicates distinct but overlapping binding behaviours by IRP1A and IRP1B to histone proteins. To examine this further, we carried out genome-wide transcript profiling of hand-dissected ring glands (which contain the PG) that expressed one of six Flag-tagged transgenes in a PG-specific manner: *IRP1A^WT^* and *IRP1B^WT^*, which are both wild type; *IRP1A^C450S^* and *IRP1B^C447S^*, both of which can only assume the apo-form and are predominantly cytoplasmic; as well as *IRP1A^3R3Q^* and *IRP1B^3R3Q^*, both of which are presumed to be non-RNA-binding and can enter nuclei (not shown) (Supplementary Table 1). The design of this approach was based on the idea that the transcriptional changes elicited by IRP1A^3R3Q^ and IRP1B^3R3Q^ should largely result from their nuclear function, as they are predicted to have lost RNA-binding capability. When we examined the 234 most significantly downregulated genes by IRP1A^3R3Q^, we noticed strong enrichment of genes involved in iron-dependent processes, most notably steroid hormone biosynthesis (Tables 1, 2). The results for IRP1B^3R3Q^ were very similar (Pearson correlation 0.896, P < 0.001), and will not be discussed separately here.

Specifically, six of the seven known Halloween enzymes were found among the 60 most strongly downregulated genes. Furthermore, other genes involved in ecdysone biosynthesis, such as transcription factors, sterol transporters, heme biosynthesis, and iron-sulfur cluster assembly proteins were also significantly enriched in this set (Table 2). Remarkably, the fold changes for these genes were highly consistent with the predicted functions of these IRP1A variants. In particular, wild type IRP1A displayed the same trend as IRP1A^3R3Q^, but fold changes were less severe. This is consistent with the idea that wild type IRP1A is still capable of binding to mRNAs, effectively reducing nuclear IRP1A levels, resulting in similar, but reduced responses. IRP1A^C450S^ is mostly cytoplasmic, but interacted weakly with histones (Supplementary Data 1), suggesting some nuclear presence. However, most gene expression changes were not significant, suggesting that IRP1A^C450S^ had little influence on altering the expression profiles of this gene set. In conclusion, the use of different IRP1A variants allowed us to distinguish the different subcellular roles of IRP1A, and we could show that IRP1A^3R3Q^, but not IRP1A^C450S^, dramatically and significantly altered the expression of genes involved in iron-dependent processes.

## Discussion

In this report, we demonstrated that the *Drosophila* glycogen branching enzyme, AGBE, has hitherto undiscovered and essential roles in the regulation of cellular iron homeostasis. We expect that AGBE's role in iron is not limited to the PG since genome-wide expression profiling indicates that AGBE is widely expressed[29]. While AGBE has not been directly linked to iron homeostasis, a possible indirect link exists because mutations in *RBCK1* (RanBP-type and C3HC4-type zinc finger-containing protein 1), a gene that encodes an E3 ubiquitin ligase, cause Polyglucosan Body Myopathy, a recently described glycogen storage disorder[30]. Intriguingly, RBCK1 was shown to control cellular iron homeostasis by degrading the oxidized form of IRP2[31], raising the idea that glycogen and iron processes are linked on multiple levels.

The finding that AGBE regulates cellular iron homeostasis led to another surprising discovery, namely that IRP1, in a tissue-specific manner, enters nuclei in its holoform to transcriptionally downregulate iron-intensive processes. Further, both AGBE and IRP1A interact with Cisd2, a close homolog of vertebrate mito-NEET, which is known to repair oxidatively damaged IRP1. We conclude that the glycogen metabolism enzyme AGBE has a "moonlighting" function in aiding Cisd2 in this repair process, and that loss of either Cisd2- or AGBE-function results in the accumulation of damaged IRP1A, which interferes with nuclear entry (Figs. 3f, 4j, 5) and IRP1A aconitase activity (Supplementary Fig. 7C). This is consistent with our finding that only holo-IRP1A can translocate to nuclei since both BPS-treatment and a mutation in a critical cysteine required for Fe–S binding (IRP1A^C450S^ and IRP1B^C447S^) impairs nuclear access (Fig. 3d).

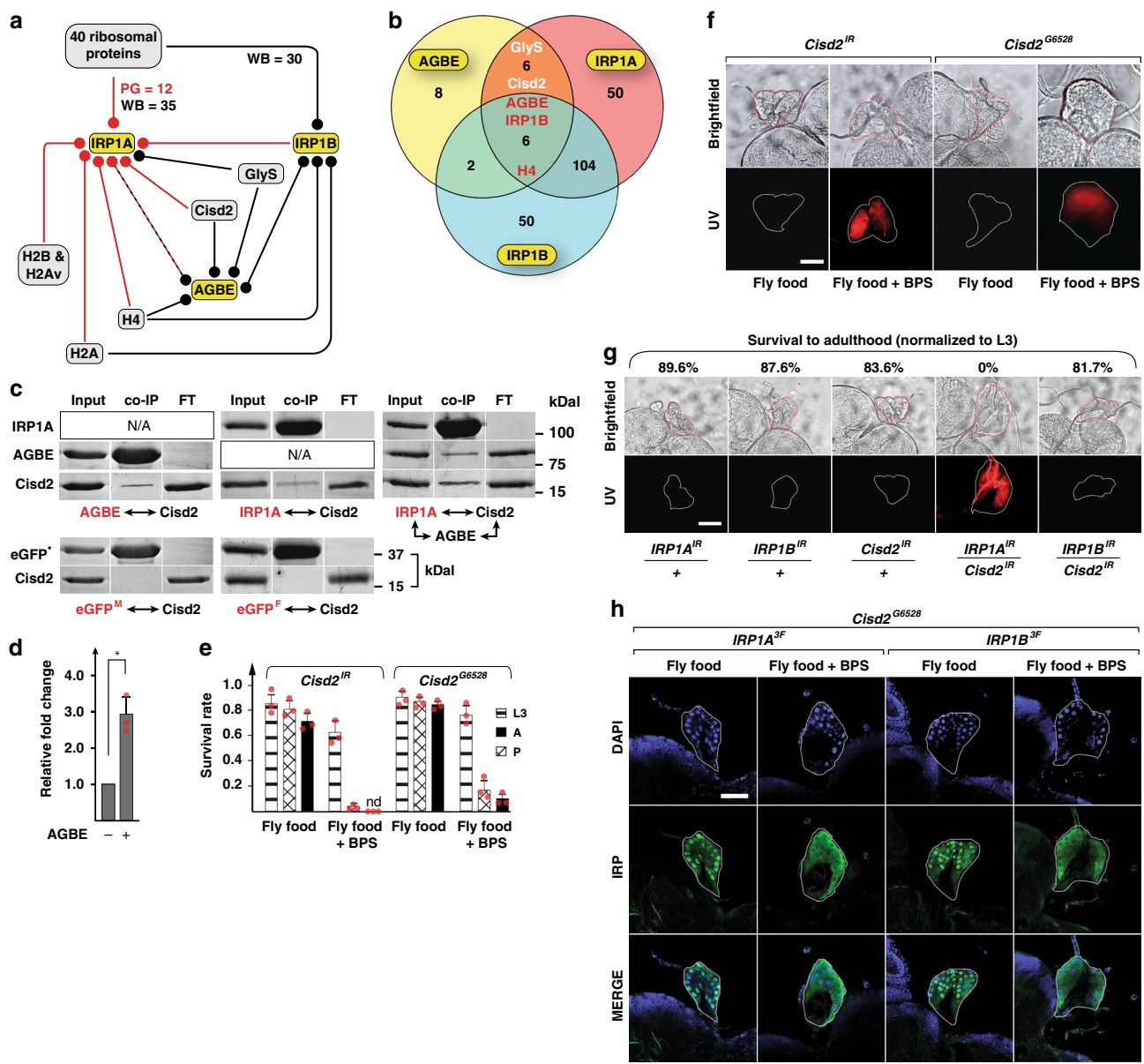

**Fig. 4** Cisd2 interacts with IRP1A and AGBE. **a** Protein–protein interaction map. Lines carrying knock-in alleles encoding Flag-tagged AGBE, IRP1A and IRP1B (yellow boxes, Supplementary Fig. 4) were used to produce bait (circle) for immunoprecipitation followed by mass spectrometry (MS) to identify physically bound proteins to the bait. Whole-body (WB, black) and prothoracic gland samples (PG) were used. Red: detected in both PG and WB samples. Dashed line: Only WB samples were tested for AGBE. H2Av, H2A, H2B and H4 are histone proteins. GlyS = Glycogen Synthase. **b** Venn diagram depicting overlaps of immunoprecipitated proteins from endogenously tagged proteins (WB samples). H4 & GlyS see A. **c** Co-transfection of Schneider 2 cells with plasmids encoding Myc-tagged AGBE, Flag-tagged IRP1A and HA-tagged Cisd2, followed by immunoprecipitation via anti-Myc or anti-Flag antibodies and Western blotting. Names shown in red indicate the protein used as bait. Myc-tagged and Flag-tagged enhanced GFP (eGFP$^M$ and eGFP$^F$, respectively) served as negative controls. Presence of co-immunoprecipitated proteins were tested with anti-HA antibodies and anti-Myc antibodies. **d** Quantification of immunoprecipitated Cisd2 in the triple co-transfection experiment shown above in C. Graph shows relative fold change of co-immunoprecipitated Cisd2 with Flag-IRP1A as bait in the presence or absence of AGBE. Data was normalized to the amount of Cisd2 protein in the absence of co-transfected AGBE. The asterisk indicates a *P*-value < 0.05 according to the Student's *t* test. Error bars represent standard deviation based on three biological replicates. **e** Survival rates of Cisd2$^{IR}$-RNAi animals and Cisd2$^{G6528}$ mutants on fly food ± BPS. nd = not detected. Error bars represent standard deviation from three biological replicates (each sample contained 50 individuals). **f** Autofluorescence/protoporphyrin accumulation in prothoracic glands (PG) of *PG > Cisd2$^{IR}$* and *Cisd2$^{G6528}$* larvae reared on fly food ± BPS. Scale bar = 250 μm. **g** Genetic interaction between *Cisd2* and *IRP1A* on regular (=iron-replete) fly food based on autofluorescing PGs and survival of the corresponding RNAi lines. All lines express RNAi via a PG-specific Gal4 driver (*phm22-Gal4 = PG >*). Scale bar = 250 μm. **h**. Subcellular localization of Flag-tagged IRP1A and IRP1B proteins expressed from knock-in alleles (Supplementary Table 1) in *Cisd2$^{G6528}$* mutants reared on fly food ± BPS. For control larvae, see Supplementary Fig. 10. Scale bar = 500 μm. Source data are provided as a Source Data file.

Thus, cells possess two mechanisms by which functional IRP1A can be generated. One is by "de novo" insertion via the Cytosolic Iron-sulfur protein Assembly (CIA) machinery, a highly conserved machinery that assembles and inserts [4Fe-4S] clusters

into client proteins[32]. Once inserted, cells require a second "maintenance" process via the mitoNEET/AGBE proteins to replace oxidatively damaged clusters with functional units (Fig. 5). This elegantly explains as to why *AGBE*-loss-of-function

**Table 1 Transcriptional responses of expressing *IRP1A* alleles in the prothoracic gland.**

| Rank (n/234) | Symbol | Description | FC 3R3Q:C | P | FC IRP1A:C | P | FC C450S:C | P |
|---|---|---|---|---|---|---|---|---|
| 6 | *dib* | Ecdysone biosynthesis/heme binding/P450 | −13.7 | 4.0E-03 | −2.4 | >0.05 | −1.7 | >0.05 |
| 7 | *phm* | Ecdysone biosynthesis/heme binding/P450 | −11.7 | 1.7E-02 | −2.2 | >0.05 | 1.1 | >0.05 |
| 12 | *sad* | Ecdysone biosynthesis/heme binding/P450 | −9.7 | 3.2E-02 | −1.7 | 1.7E-02 | 1.2 | >0.05 |
| 13 | *Start1* | Sterol transport | −9.6 | 4.7E-03 | −3.1 | 3.3E-02 | −2.3 | >0.05 |
| 15 | *CG7322* | Short-chain dehydrogenase | −9.0 | 1.4E-02 | −1.8 | >0.05 | −1.1 | >0.05 |
| 23 | *Cyp6g2* | Heme binding/P450 | −7.2 | 1.7E-02 | −1.1 | >0.05 | −1.2 | >0.05 |
| 26 | *spidey* | Short-chain dehydrogenase | −7.0 | 1.8E-02 | −1.6 | 2.9E-02 | −1.6 | 1.6E-02 |
| 34 | *nvd* | Ecdysone biosynthesis/iron sulfur cluster | −6.1 | 1.7E-02 | −2.1 | 8.9E-03 | −1.6 | >0.05 |
| 35 | *sro* | Ecdysone biosynthesis/short-chain dehydrogenase | −6.0 | 2.1E-03 | −4.2 | 1.3E-02 | −1.2 | >0.05 |
| 41 | *GstE14* | Ecdysone biosynthesis/glutathione S transferase | −5.6 | 2.8E-02 | −2.6 | >0.05 | −1.4 | >0.05 |
| 44 | *ND-15* | NADH:ubiquinone oxidoreductase, iron-sulfur subunit 5 | −5.2 | 2.8E-02 | −3.3 | >0.05 | −2.7 | 8.9E-03 |
| 47 | *ouib* | Ecdysone biosynthesis/zinc finger | −4.9 | 2.4E-02 | −2.1 | >0.05 | −1.8 | >0.05 |
| 51 | *spok* | Ecdysone biosynthesis/heme binding/P450 | −4.8 | 3.6E-03 | −2.4 | 2.9E-03 | −1.2 | >0.05 |
| 52 | *scu* | Short-chain dehydrogenase/reductase | −4.7 | 1.1E-02 | −1.3 | >0.05 | 1.2 | >0.05 |
| 55 | *Cyt-b5* | Cytochrome b5-like heme/steroid binding domain | −4.5 | 4.4E-02 | −1.8 | >0.05 | −1.4 | 1.0E-02 |
| 56 | *CG17928* | Cytochrome b5-like heme/steroid binding domain | −4.5 | 1.4E-02 | −1.8 | >0.05 | −2.3 | 3.0E-03 |
| 64 | *Tig* | Heme oxygenase-like | −4.2 | 6.0E-03 | −1.7 | >0.05 | −1.2 | >0.05 |
| 74 | *ND-19* | NADH:ubiquinone oxidoreductase | −3.9 | 4.0E-02 | −1.5 | >0.05 | −1.2 | >0.05 |
| 98 | *Fdx2* | Ferredoxin/iron sulfur cluster assembly | −3.4 | 1.3E-03 | −1.6 | >0.05 | 1.3 | >0.05 |
| 107 | *Npc2a* | Sterol transport | −3.3 | 8.4E-03 | −1.9 | >0.05 | −1.6 | >0.05 |
| 116 | *Pbgs* | Heme biosynthesis | −3.2 | 6.4E-03 | −1.5 | >0.05 | −1.2 | >0.05 |
| 125 | *Npc1a* | Sterol transport | −3.1 | 3.2E-04 | −2.3 | 2.4E-02 | −1.2 | >0.05 |
| 146 | *Drat* | Response to hypoxia | −2.9 | 1.6E-02 | −1.4 | >0.05 | −1.6 | 1.1E-02 |
| 170 | *CG31548* | Short-chain dehydrogenase/reductase | −2.7 | 3.1E-02 | −1.8 | >0.05 | −1.4 | >0.05 |
| 173 | *CG32857* | Nfu1 homolog/iron sulfur cluster assembly | −2.7 | 3.7E-02 | −1.2 | >0.05 | −1.2 | >0.05 |
| 184 | *ance* | Ecdysone biosynthesis/zinc finger | −2.7 | 3.7E-02 | −2.0 | >0.05 | −1.2 | >0.05 |
| 202 | *Alas* | Heme biosynthesis | −2.6 | 1.5E-02 | −1.0 | >0.05 | 12.9 | 2.6E-04 |
| 212 | *CG12056* | Cytochrome b5-like heme/steroid binding domain | −2.6 | 1.7E-02 | −1.5 | >0.05 | 1.1 | >0.05 |
| 218 | *CG2254* | Short-chain dehydrogenase/reductase | −2.6 | 6.7E-03 | 1.2 | 5.1E-05 | −1.1 | >0.05 |
| 221 | *Vhl* | Response to hypoxia | −2.6 | 2.0E-02 | −1.6 | >0.05 | 1.1 | >0.05 |

RNA-Seq analysis of prothoracic gland (PG) samples with PG-specific expression of *IRP1A^3R3Q* (non-RNA-binding and nuclear), wild type *IRP1A* and *IRP1A^C450S* (constitutively RNA-binding and largely cytoplasmic) transgenes (Supplementary Table 1). A total of 234 genes were significantly downregulated by *IRP1A^3R3Q* expression, using a cutoff of −2.5-fold and P < 0.05. The fold changes (relative to control = C) and P-values (t-test) are shown for all three conditions and sorted by relative fold changes of IRP1A^3R3Q vs. controls. For all 234 genes, see Supplementary Data 3, for term enrichment in this set, see Table 2 and Supplementary Data 4

animals can only be rescued by expressing a wild type *IRP1A* transgene, but not by the constitutively RNA-binding form (IRP1A^C450S): Sustained transgenic expression of wild type *IRP1A* allows cells to produce sufficient functional IRP1A before oxidative damage occurs, simply because the CIA machinery is able to maintain critical levels of holo-IRP1A, despite the absence of a functioning AGBE/mitoNEET repair machinery (Fig. 5). In contrast, IRP1A^C450S fails to rescue, since it cannot assume the holoform that is required to enter nuclei.

What could be the biological context that requires IRP1A and IRP1B entering the nucleus? In the PG, iron demands are not only exceedingly high, but they also must change dramatically as the need for Halloween enzyme production changes during development (Supplementary Fig. 1 and Fig. 5). It is therefore plausible that once production of ecdysone has peaked, PG cells need to downregulate all processes that are tied to the synthesis of steroids. Since all but one of the ecdysone-producing Halloween enzymes require iron co-factors, it is necessary to downregulate iron-cofactor production in concert with the proteins that require them. We hypothesize that peak levels of bioavailable iron correlate with maximal nuclear activity of holo-IRP1, resulting in a downregulation of iron-dependent processes, in particular,

steroid hormone biosynthesis. As such, holo-IRP1 appears to have a hitherto undescribed role in iron regulation: Apo-IRP1, as a cytoplasmic mRNA-binding protein, responds to a drop in cellular iron and facilitates an increase of bioavailable iron, yet nuclear holo-IRP1 transcriptionally downregulates iron- and heme-dependent processes once peak iron demand is over (Fig. 5).

How does IRP1 cause the coordinated transcriptional downregulation of iron-dependent processes? An attractive model is that IRP1 proteins interact with a subset of modified histone tails, rather than binding to their histone partners in a non-discriminate fashion. We see two possible scenarios from here. First, histone-bound IRP1 could directly recruit repressive chromatin factors such as histone deacetylases or chromatin remodellers and simply act as a co-factor that serves as a readout for cellular iron concentrations. The second and perhaps more intriguing possibility is that holo-IRP1 controls nuclear citrate levels via its aconitase function to indirectly regulate histone acetylation. Nuclear citrate is converted to acetyl-CoA and oxaloacetate by nuclear ATP-citrate lyase (ACL), a metabolic enzyme with critical roles in histone acetylation[33]. Acetyl-CoA is the principal substrate for histone acetylation and is considered a

**Table 2 Term enrichment analysis via DAVID tools.**

| Category | Term | P | E |
|---|---|---|---|
| KEGG | Insect hormone biosynthesis | 9.9E-07 | 17.8 |
| Keywords | Oxidoreductase | 2.9E-06 | 2.8 |
| Biological Process | Ecdysone biosynthetic process | 5.1E-05 | 22.5 |
| InterPro | NAD(P)-binding domain | 2.5E-04 | 3.9 |
| Cellular Component | Mitochondrion | 4.0E-03 | 2.1 |
| InterPro | Short-chain dehydrogenase/reductase, conserved site | 9.0E-03 | 9.1 |
| Keywords | Iron | 4.0E-02 | 2.5 |
| InterPro | Cytochrome b5-like heme/steroid binding domain | 4.0E-02 | 9.3 |
| Molecular Function | Iron-sulfur cluster binding | 4.0E-02 | 8.9 |
| Molecular Function | Heme binding | 4.6E-02 | 2.7 |
| Keywords | Metalloprotease | 6.2E-02 | 4.4 |
| Biological Process | Positive regulation of ecdysteroid biosynthetic process | 9.8E-02 | 19.5 |

We analyzed 234 genes (see Table 1) for GO terms (Biological Process, Molecular Function, Cellular Component), keywords, Kegg pathway terms, and InterPro protein domains. P = P-value (based on EASE Score, a modified Fisher Exact P-Value), E = fold enrichment. For full GO results, see Supplementary Data 4

**Fig. 5** Model for AGBE-mitoNEET/Cisd2-IRP1A function in tissues with dynamic iron requirements. In *Drosophila*, iron demand peaks prior to maximal ecdysone production to equip ecdysone-synthesizing enzymes with iron cofactors. Newly synthesized IRP1A receives Fe–S clusters (red circles) from the Cytosolic Iron-sulfur cluster Assembly (CIA) machinery, which produces [4Fe-4S] sulfur clusters from an unidentified mitochondrial precursor molecule, X-S (produced in mitochondria by ISC = Iron-Sulfur Cluster Assembly Machinery). Oxidatively damaged IRP1A (IRP1A-HOLO*) requires the mitoNeet/ Cisd2 and AGBE proteins to replace impaired clusters with functional units. Holo-IRP1A is both needed for the aconitase function as well as nuclear entry. Loss of AGBE or mitoNEET/Cisd2 function results in a depletion of holo-IRP1A, and the concomitant loss of nuclear IRP1A, explaining why *AGBE* mutants cannot be rescued with the IRP1A$^{C450S}$ form, which only assumes the apo-form since it cannot incorporate an Fe–S cluster. The model suggests that once iron demand has peaked and nuclear holo-IRP1A levels become maximal, the protein acts to throttle expression of genes acting in steroid, heme and iron metabolism in anticipation of falling iron demands. As such, IRP1A has two functions: 1. As known from mammalian cells it acts in response to low cellular iron levels as an RNA-binding protein that promotes increased iron availability, and 2. as a new function, it responds to peak iron levels as a nuclear protein to promote downregulation of processes depending on iron and heme (TR = Transcriptional Regulation).

highly regulated nuclear metabolite that controls histone acetylation status[34,35]. Histone-bound IRP1A and IRP1B could then act by converting citrate into isocitrate and deplete Acetyl-CoA levels, thus negatively impacting gene expression by promoting histone de-acetylation.

Mutations in human GBE1 cause Andersen disease, also known as Glycogen Storage Disease Type IV (GSD IV)[36], but the gene has not been linked to iron homeostasis yet. There are strong indications, however, that GBE1 has hitherto undocumented roles in vertebrate iron metabolism as well. Besides the earlier mentioned interaction with IRP1, GBE1 was identified by whole-exome sequencing as a novel mitochondrial disorder locus[37], consistent with a study that found abnormal mitochondria in GSD IV patients[38]. Furthermore, GBE1 is transcriptionally upregulated in response to hypoxia and one of the most strongly induced genes upon nickel exposure[39–41]. Nickel exposure elicits hypoxic responses, and at least in vertebrates, hypoxia and iron metabolism are tightly linked[42]. Perhaps most intriguingly, Nrf2, a transcription factor controlling mitochondrial biogenesis and important iron metabolism genes, was shown to bind directly to

the GBE1 promoter[43,44], raising the idea that this glycogen enzyme is coordinately controlled with other key iron genes. Taken together, our findings strongly suggest that the disease etiology of GSD IV needs to be re-assessed from the perspective that GBE1 has a key role in cellular iron homeostasis, and that there must be a re-evaluation of current therapeutic strategies in the future.

The loss-of-function phenotypes for Cisd2/mitoNEET, AGBE, and IRP1A are very similar since they display protoporphyrin accumulation that disappears under iron-replete conditions. This is consistent with the idea that depleting Cisd2/mitoNEET or AGBE equates the loss of IRP1A function, as both appear to act in concert to replace damaged Fe–S clusters in IRP1A. We have shown that IRP1A is an essential protein required for responding to low dietary iron levels, but IRP1A null mutants survive on an iron-rich diet. Therefore, our data strongly suggests that Cisd2/AGBE are gatekeepers that ensure proper functioning of IRP1A, a function that becomes non-essential in iron-replete conditions. Vertebrates encode three mitoNEET-like proteins, Cisd1-3[23]. Drosophila lacks a direct Cisd1 orthologue but harbours copies of Cisd2 and CG3420 (Cisd3). Of the two, Cisd2 is more similar to Naf-1 and mitoNEET[45]. We were unable to identify any defects when disrupting CG3420 function via RNAi (not shown), suggesting that Cisd2 is the functional equivalent of mitoNEET in Drosophila. The fact that a) fly Cisd2 interacts physically and genetically with IRP1A and b) that mutations in either gene resulted in comparable phenotypes strongly supports the notion that IRP1A function depends on Cisd2, consistent with the finding in vertebrates that mitoNEET is involved in repairing oxidatively damaged Fe–S clusters. Similar to Drosophila IRP1A, null mutations of mouse IRP1 or IRP2 are non-lethal under normal conditions, however, the double knockout is embryonic lethal[46–50]. IRP1 null mutants exhibit increased blood haemoglobin levels (polycythemia)[46,48,50] and one lab reported[48] that these mice developed also pulmonary hypertension that was exacerbated by exposure to a low iron diet, causing premature death.

The existing parallels between vertebrate IRP1 and Drosophila IRP1A raise the interesting question as to whether vertebrate IRP1 has a nuclear role as well, and whether it is conceivable that such a function has been hitherto overlooked. Consistent with this idea, a search of a human protein–protein interaction database[15] found that IRP1 interacts with Histone 2Ab[51] (out of 19 reported proteins in total). In addition, we found that the presence of IRP1A in nuclei varies with tissue and nutritional conditions, raising the possibility that nuclear translocation occurs only under certain circumstances. This may be controlled by physiological parameters, depending on whether a tissue has high or normal iron requirements, and may be temporally regulated during development, as is the case for the PG. Further, we showed that the vertebrate IRP proteins use the same principles as their Drosophila counterparts for nuclear entry since only human IRP1 has the ability to translocate to Drosophila nuclei, while IRP2, which lacks an Fe–S cluster, does not. It should also be noted that our findings were aided by the fact that the tissues we investigated are polytene, and consequently harbour, compared to most human cells, very large nuclei that allow easy visualization of nuclear proteins. Finally, we searched the literature for studies that had examined the subcellular localisation of IRP1 in more detail. To the best of our knowledge, the existing data relies solely on cell culture experiments with SW1088 and HepG2 cells, which reported IRP1 to mainly reside in the cytosol, but also found IRP1 associated with the endoplasmic reticulum and the Golgi apparatus[52,53]. While the effects of hypoxic and iron-deprived conditions on IRP1 localisation were tested, iron-rich conditions were not. Taken together, we believe that IRP1, at least in certain circumstances, behaves like its Drosophila counterpart, and enters nuclei where it is physiologically relevant. Future studies will have to revisit this issue in vertebrates.

## Methods

**Drosophila stocks and husbandry.** We obtained the following stocks from the Bloomington Drosophila Stock Center: $w^{1118}$ (#3605), UAS-AGBE$^{IR2}$ (#42753), Cisd2$^{G6528}$ (#30170), Tubulin-Gal4/TM3 Ser.GFP (#5138), UAS-eGFP (#5431), UAS-FLP (#4539), UAS-CD8.Venus (#65609), Vas.Cas9 (#51323). Stocks UAS-AGBE$^{IR1}$ (#108087), UAS-IRP1A-RNAi (#105583), UAS-IRP1B-RNAi (#110637), UAS-Cisd2-RNAi (#104501) were obtained from the Vienna Drosophila Resource Center. We used CRISPR/CAS9 to generate the following knock-in and knock-out alleles (see Supplementary Fig. 4 for details): AGBE$^{FCF}$, AGBE$^{FCM}$, IRP1A$^{3F}$, IRP1A$^{C450S.3F}$, IRP1A$^{3R3Q.3F}$, IRP1B$^{3F}$, IRP1A$^{FCF}$/TM3 Ser.GFP, IRP1A$^{KO}$/TM3 Ser.GFP, IRP1B$^{KO/KO}$. We also generated transgenic lines based on the PhiC31 system: UAS-3xFlag-IRP1A$^{WT}$, UAS-IRP1A$^{WT}$, UAS-3xFlag-IRP1A$^{C450S}$, UAS-IRP1A$^{C450S}$, UAS-3xFlag-IRP1A$^{3R3Q}$, UAS-3xFlag-IRP1B$^{WT}$, UAS-IRP1B$^{WT}$, UAS-3xFlag-IRP1B$^{C447S}$, UAS-IRP1B$^{C447S}$, UAS-3xFlag-IRP1B$^{3R3Q}$, UAS-Yeast Aco1$^{WT}$, UAS-Yeast Aco1$^{\Delta Sp}$, UAS-3xFlag-hIRP1$^{WT}$, UAS-3xFlag-hIRP2$^{WT}$, dU6-3-IRP1A$^{gRNA}$ (Supplementary Table 1). $y^1 w* P[^{nos-PhiC31.NLS}]X$; P[carryP]attP40(II) and $y^1 w* P[nos-PhiC31/int.NLS]X$; P[carryP]attP2(III) were gifts from BestGene Inc. Phm22-Gal4 was a kind gift from Michael O'Connor's lab. Stocks were maintained on a standard cornmeal diet unless otherwise specified.

**Survival studies.** Regular fly food refers to "NutriFly"-based media, which follows the standard recipe from the Bloomington Drosophila Stock Center (https://bdsc. indiana.edu/information/recipes/bloomfood.html). Flies were reared at 25 °C and 60–70% humidity. Prior to any fly-based experiments, stocks were reared on NutriFly media for at least two generations. Modified media were prepared by adding compounds (e.g., iron) during the preparation process. For iron-enriched media, a 1 M stock solution of Ferric Ammonium Citrate (FAC) (Sigma #F5879) was used to make NutriFly containing 1 mM FAC. For iron-depletion media, Nutrifly containing 100 µM BPS (Sigma #146617) was prepared. For egg collections, flies were allowed to lay eggs for 3× 1 h in order to reduce egg retention and minimize the presence of old embryos. For each vial, 50 embryos were then collected in 1-h intervals. Embryos were counted and transferred to vials containing appropriate media. Larval survival was scored for every stage. At least three independent crosses (=three biological replicates) were carried out per experimental condition.

**Construction of transgenic lines.** For transgene properties, see Supplementary Table 1. cDNAs were obtained from the Drosophila Genomic Resource Center (AGBE: #RE12027, IRP1A: #LD36161, IRP1B: #LD13178, Cisd2: #RE49709). Human IRP1 cDNA (#HG10966-UT) and human GBE1 cDNA (#HG18919-UT) were acquired from Sino Biological Inc, while the human IRP2 cDNA (#OHS1770-202318020) was obtained from Dharmacon. To generate equivalent expression of transgenic constructs, we used PhiC31 vectors pBID-UASC-FG (Addgene #35201) and pBID-UASC-G (Addgene #35202) to ensure insertion into the same locus[54]. Vector backbones were amplified via PCR to generate two fragments per vector and fused to cDNA fragments via the Gibson reaction. Mutations were generated via Q5 mutagenesis PCR (NEB #M0491S) following the manufacturer's instructions. Fused fragments were cloned into DH5α E. coli competent cells, and validated by Sanger sequencing. For primers see Table 3.

**Generation of CRISPR/CAS9 fly lines.** We identified optimal target gRNA sites by relying on comparable results from two independent programs, "CRISPR Optimal Target Finder" (University of Wisconsin; http://tools.flycrispr.molbio. wisc.edu/targetFinder/index.php) and Harvard's "Find CRISPR" sgRNA design tool (http://www.flyrnai.org/crispr/index.html)[55]. Target sites were confirmed by sequencing corresponding loci in the Vas.Cas9 line (Bloomington #51323) that we used for embryo injections. CRISPR lines were generated via CRISPR/Cas9 homology-directed repair to replace endogenous alleles. Plasmids carrying gRNA target sites were cloned into pCFD3 (Addgene #49410) for AGBE$^{FCF}$, AGBE$^{FCM}$, IRP1A$^{3F}$ and IRP1B$^{3F}$ constructs, or pCFD5[56,57] (Addgene #73914) for IRP1A$^{KO}$, IRP1A$^{FCF}$ and IRP1B$^{KO}$. All donor template fragments were amplified from genomic DNA via PCR and cloned into the pDsRed-attP vector (Addgene #51019)[57]. For primers see Table 3.

**Embryo injection.** PhiC31 constructs were injected at 500–600 ng/µl concentrations, while CRISPR plasmids were used at a concentration of 100–150 ng/µl for the double gRNA plasmid and 500–600 ng/µl for the donor template. Injections were performed either at the University of Alberta or via GenetiVision Corporation using standard procedures. 300–500 embryos were injected per construct. Surviving adults were backcrossed to $w^{1118}$ and used to generate independent lines.

**Immunostaining.** Brain-ring gland complexes (BRGC) were isolated from (unless stated otherwise) 40–42 h 3rd instar larvae and transferred to 1× PBS. Samples were fixed in 1× PBS 4% formaldehyde (ThermoFisher #28906) for 20 min at room temperature (RT) followed by washing in 1× PBS 0.3% Triton (Sigma #T9284)

**Table 3 Primer sequences.**

| Primer name | Primer sequence (5′-3′) |
| --- | --- |
| Generation of trangenic cDNA lines | |
| attB1 IRP1A FP | CAAGTTTGTACAAAAAAGCAGGCTATGTCCGGCTCCGGCGCCAATC |
| attB2 IRP1A RP | TCCACTTTGTACAAGAAAGCTGGGTCTAATCCAGCATTTTGCGTATC |
| attB1 IRP1B FP | TCAAGTTTGTACAAAAAAGCAGGCTATGTCAGGCGCCAATCCCTTC |
| attB2 IRP1B RP | CCCACTTTGTACAAGAAAGCTGGGTTTAAGAGAGCATTTTGCGAATCATG |
| attB1 yeast Aco1 DelSp FP | CCAAGTTTGTACAAAAAAGCAGGCTGTCTCCAACTTGACTAGAGATTC |
| attB2 yeast Aco1 RP | CACTTTGTACAAGAAAGCTGGGTTTTCTTCTCATCGGCCTTAATTTTATTTAAG |
| attB1 yeast Aco1 WT FP | CCAAGTTTGTACAAAAAAGCAGGCTATGCTGTCTGCACGTTCTG |
| attB1 human IRP1 FP | CAAGTTTGTACAAAAAAGCAGGCTATGAGCAACCCATTCGCAC |
| attB2 human IRP1 RP | CCCACTTTGTACAAGAAAGCTGGGTCTACTTGGCCATCTTGCGGATC |
| attB1 human IRP2 FP | TCAAGTTTGTACAAAAAAGCAGGCTATGGACGCCCCAAAAGCAG |
| attB2 human IRP2 RP | TCCACTTTGTACAAGAAAGCTGGGTCTATGAGAATTTTCGTGCCAC |
| attB1 FG RP | AGCCTGCTTTTTTGTACAAACTTGATACCGGTGCTTGTCATCGTC |
| miniwhite FP | GAGTTCGATGTGTTTATTAAGGGTATCTAGCATTAC |
| miniwhite RP | GTAATGCTAGATACCCTTAATAAACACATCGAACTC |
| attB2 FG FP | ACCCAGCTTTCTTGTACAAAGTGGAGACGTAAGCTAGAGGATCTTTGTG |
| attB1 UASCG RP | AGCCTGCTTTTTTGTACAAACTTGAGATATCGAGCTCTCCCGGGAATTCGGATC |
| attB2 UASCG FP | ACCCAGCTTTCTTGTACAAAGTGGAGATATCGCATGCGGTACCTC |
| Yeast Aco1 WT RevMut FP | ATGCTGTCTGCACGTTCTGCCATCAAGAGACCCATTGTTCGTGGTCTTGCGACAGTCTCCAACTTGACTAGAGATTC |
| Mutagenesis of *IRP1A* variants | |
| IRP1A C450S Mut FP | ATCACCTCGAGCACGAACACTTC |
| IRP1A C450S Mut RP | GGCGGCAATCACAAAGATC |
| IRP1A R549Q Mut FP | CGGGCAATCAGAATTTCGAG |
| IRP1A R549Q Mut RP | ACAGGACGCCACAGCAAAC |
| IRP1A R554Q Mut FP | ATACTAGGGCCAATTATCTGGCCAG |
| IRP1A R554Q Mut RP | TGGGATGGATCTGACCCTC |
| IRP1A R712Q Mut FP | CTATTTGTCGGAACAGGGTCTAACGCCGCGCGAC |
| IRP1A R712Q Mut RP | CGCGCTGCCGGTGACTTTCGTG |
| IRP1A R793Q Mut FP | TGGCAGCTCACAGGATTGGGCCGCCAAG |
| IRP1A R793Q Mut RP | CTGCCGTAGTCCTTGCCTAC |
| Mutagenesis of *IRP1B* variants | |
| IRP1B C447S Mut FP | TCACATCCAGCACGAACACATC |
| IRP1B C447S Mut RP | TGGCCGCAATGACAACAGATC |
| IRP1B R546Q Mut FP | AGTTTTGTCCGGAAACCAGAACTTCGAG |
| IRP1B R546Q Mut RP | CCAGCACACACGAGGCCGTTCTTCTCGATG |
| IRP1B R551Q Mut FP | CAACTATCTGGCCAGTCCTCTG |
| IRP1B R551Q Mut RP | GCCCTGGTGTTGGGATGGATCTGACCCTCGAAG |
| IRP1B R709Q Mut FP | TCTTGTCCGAGCAGAACATCACACCCCGTG |
| IRP1B R709Q Mut RP | ATCGGGCAGCAGGCGAGGTTCTAG |
| IRP1B R790Q Mut FP | CGGAAGCTCTCAGGATTGGGCCGCCAAG |
| IRP1B R790Q Mut RP | CTGCCGTAGTCCTTGCCTAC |
| Generation of AGBE CRISPR lines | |
| AGBE LA FP | CGCTGAAGCAGGTGGAATTCTATGGCAACAGTCGGTGGCTTCTG |
| AGBE FRT LA RP | GAAGTTCCTATACTTTCTAGAGAATAGGAACTTCGGAAACAGCTCTGCTCCACTG |
| AGBE FRT Middle FP | GAAGTTCCTATTCTCTAGAAAGTATAGGAACTTCGAGTGGCGACCTAATCTGTG |
| AGBE 3xFlag RP | AATATCATGATCCTTGTAGTCTCCGTCGTGGTCCTTATAGTCCATCTAGTCACTGACGCGGGCATAAAC |
| AGBE 3xFlag_3UTR FP | ACTACAAGGATCATGATATTGATTACAAAGACGATGACGATAAGTAGCTAGTCAGACGCAATTAAC |
| AGBE 3UTR RP | ACTACGATCGCAGGTGTGCAAAAGCAAGCCCAAATCCCTAAAATTC |
| AGBE RA FRT FP | ACTCATCAATGTATCTTAGAGTTCCTATTCTCTAGAAAGTATAGGAACTTC CATTGGCCAATAACAAAG |
| AGBE RA RP | TGCATGGAGATCTTTACTAGCATTTAGTTCTGCTCTCTTTGTTG |
| AGBE 3xMyc RP | TGCTCGAGGTCCTCCTCGGAGATGAGCTTTTGCTCAAGATCCTCTTCAGAAATAAGTTTTTGTTCTCTAGTCACTGAC GCGGGCATAAAC |
| AGBE 3xMyc_3UTR FP | TCCGAGGAGGACCTCGAGCAGAAGTTGATCAGCGAGGAAGACTTGTAGCTAGTCAGACGCAATTAAC |
| pCFD3 AGBE Left gRNA FP | GTCGGAGCAGAGCTGTTTCCGAG |
| pCFD3 AGBE Left gRNA RP | AAACTCGGAAACAGCTCTGCTCC |
| pCFD3 AGBE Right gRNA FP | GTCGGATTTGGGCTTGCTTTCAT |
| pCFD3 AGBE Right gRNA RP | AAACATGAAAGCAAGCCCAAATC |
| Generation of *IRP1A^KO^* CRISPR line | |
| IRP1A LA FP | TGTCGCCCTTCGCTGAAGCAGGTGGGTACGAGTGGGCGGGACAGAAG |
| IRP1A KO LA RP | GCACTACGATCGCAGGTGTGCATATAGTGGAATAATTTATCATTTTTGTGATTC |
| IRP1A KO RA FP | TATACGAAGTTATAGAAGAGCGCCAAACCAGTCCTGCTAAAAATGCCTAAC |
| IRP1A RA RP | GATTGACGGAAGAGCCTCGAGCGCTGGTGGTGTTGGTGATGTTGCTG |
| pCFD5 IRP1A Left gRNA FP | CGGCCCGGGTTCGATTCCCGGCCGATGCAGAAACATTTGTAAATTATAG GTTTTAGAGCTAGAAATAGCAAG |
| pCFD5 IRP1A Right gRNA RP | ATTTTAACTTGCTATTTCTAGCTCTAAAACATGGCCAAACCAGTCCTGCTTGCACCAGCCGGGAATCGAACCC |
| Generation of *IRP1A^3F^* CRISPR line | |
| IRP1A 3 F LA FP | CCCTTCGCTGAAGCAGGTGGTGACCTCGGTTTCGGGGCCCAAG |
| IRP1A 3 F LA RP | GATCCGGCTGGCGAGATGTGGTCGGTGGTCACTGAATCACCGAG |
| IRP1A 3 F Middle FP | CTCGGTGATTCAGTGACCACCGACCACATCTCGCCAGCCGGATC |
| IRP1A 3 F 3xFlag RP | CTTGTAATCAGTGTCATGATCTTTATAATCACCGTCATGGTCTTTGTAGTCATCCAGCATTTTGCGTATCATATAG |
| IRP1A 3 F 3xFlag FP | AGATCATGACATCGATTACAAGGATGACGATGACAAGTAGTTAGTGCGTTCGTTGACTTTTATATTC |
| IRP1A 3 F Middle RP | CGATCGCAGGTGTGCATAGTTAGGCATTTTTAGCAG |
| pCFD3 IRP1A 3 F Left gRNA FP | GTCGGCTGGCGAGATGTGGTCGG |
| pCFD3 IRP1A 3 F Left gRNA RP | AAACCGACCACATCTCGCCAGCC |
| pCFD3 IRP1A 3 F Right gRNA FP | GTCGAGCAGGACTGGTTTGGCCAT |
| pCFD3 IRP1A 3 F Right gRNA RP | AAACATGGCCAAACCAGTCCTGCT |
| Generation of *IRP1A^C450S.3F^* and *IRP1A^3R3Q.3F^* CRISPR lines | |
| IRP1A CRISPR Mut LA FP | TGGAATTCTTGCATGCTGCTAGCGTGACAACTTTCATGTGCTG |
| IRP1A CRISPR Mut LA RP | TGGATATCAAGTAATAAATTTAGATAATTTTTAAG |
| IRP1A CRISPR Mut Middle FP | TGGATGTGGATCATTAGATCGCTCCGAGAAGAAAATCGATATTATCCGGAAG |
| IRP1A CRISPR Mut Middle RP | GCACTACGATCGCAGGTGTGCATAGTTAGGCATTTTTAGCAGGACTG |
| IRP1A CRISPR Mut RA RP | TGTATGCTATACGAAGTTATAGAAGAGCGCCTGCTGGTGAATCATCGACAAG |

**Table 3** (continued)

| Primer name | Primer sequence (5′-3′) |
| --- | --- |
| IRP1A CRISPR Mut RA FP | GATTGACGGAAGAGCCTCGAGCTGCACCTGGTGCTGGTGGTG |
| IRP1A pCFD5 FP | GCGGCCCGGGTTCGATTCCCGGCCGATGCAAATTTATTACTTGATATCCAGTTTTAGAGCTAGAAATAGCAAG |
| IRP1A pCFD5 RP | ATTTTAACTTGCTATTTCTAGCTCTAAAACATGATTCACCAGCAGGCGTTTGCACCAGCCGGGAATCGAAC |
| Generation of $IRP1A^{FCF}$ CRISPR line | |
| IRP1A FCF FRT LA RP | GAAGTTCCTATACTTTCTAGAGAATAGGAACTTCTAGTGGAATAATTTATCATTTTTG |
| IRP1A FCF FRT Middle FP | GAAGTTCCTATTCTCTAGAAAGTATAGGAACTTCTAATTTACAAATGTTTCATTTTAAG |
| IRP1A FCF FRT Middle RP | GCACTACGATCGCAGGTGTGCATAGAAGTTCCTATACTTTCTAGAGAATAGGAACTTCCATCGGCATTTCTGCTATC |
| Generation of $IRP1A^{gRNA}$ CRISPR line | |
| IRP1A gRNA FP | GCGGCCCGGGTTCGATTCCCGGCCGATGCCTTTATCCGGATAGCGTTGT GTTTTAGAGCTAGAAATAGCAAG |
| IRP1A gRNA FP | ATTTTAACTTGCTATTTCTAGCTCTAAAACCCCAGCTCGCGGACAGCATCTGCACCAGCCGGGAATCGAACCC |
| Generation of $IRP1^{BKO}$ CRISPR line | |
| IRP1B KO LA FP | TCGCCCTTCGCTGAAGCAGGTCACAGCAGACAGTTAATAC |
| IRP1B KO LA RP | TACGATCGCAGGTGTGCATAAGTAATCGACAGAGCTCGTGCAATC |
| IRP1B KO RA FP | ACGAAGTTATAGAAGAGCAGGGTGGCTTCCGCAAACGAATTG |
| IRP1B KO RA RP | CTTATGCATGGAGATCTTTACTAGCGTAGAGCATCTGCACCAGATTTCG |
| pCFD5 IRP1B Left gRNA FP | CGGCCCGGGTTCGATTCCCGGCCGATGCCAGAGCTCTGTCGATTACTGATGTTTTAGAGCTAGAAATAGCAAG |
| pCFD5 IRP1B Right gRNA RP | ATTTTAACTTGCTATTTCTAGCTCTAAAACCCCGTGGCCCCACCGCAACCTGCACCAGCCGGGAATCGAACCC |
| Generation of $IRP1^{B3F}$ CRISPR line | |
| IRP1B 3 F LA FP | TGGAATTCTTGCATGCTAGCACTTCCCCATCGATGAGAATACTC |
| IRP1B 3 F LA RP | GATGTCCTGCAAGAAACACATTCTTGCCATTG |
| IRP1B 3 F Middle FP | CAATGGCAAGAATGTGTTCTTGCAGGACATC |
| IRP1B 3 F 3xFlag RP | TCCTTGTAATCGATGTCATGATCTTTATAATCACCGTCATGGTCTTTGTAGTCAGAGAGCATTTTGCGAATCATGTAGTTG |
| IRP1B 3 F 3xFlag FP | TATAAAGATCATGACATCGATTACAAGGATGACGATGACAAGTAAGCAACTCATCTTATTTTG |
| IRP1B 3 F Middle RP | ACGATCGCAGGTGTGCATAGTGGCCCCACCGCAACCCCTTAAG |
| pCFD3 IRP1B 3 F Left gRNA FP | GTCGATGTGTTCTTGCAGGACATC |
| pCFD3 IRP1B 3 F Left gRNA RP | AAACGATGTCCTGCAAGAACACAT |
| pCFD3 IRP1B 3F Right gRNA FP | GTCGGTTGCGGTGGGGCCACGGG |
| pCFD3 IRP1B 3F Right gRNA RP | AAACCCGTGGCCCCACCGCAACC |
| Generation of S2 cells transfection constructs | |
| attB1 eGFP FP | CAAGTTTGTACAAAAAAGCAGGCTATGGTGAGCAAGGGCGAGGAGCTGTTC |
| attB2 eGFP no stop codon RP | CACTTTGTACAAGAAAGCTGGGTCTTGTACAGCTCGTCCATGCCGAG |
| attB1 AGBE FP | CCAAGTTTGTACAAAAAAGCAGGCTATGGCCGAGGCTAAGGACATC |
| attB2 AGBE no stop codon RP | CACTTTGTACAAGAAAGCTGGGTGTCACTGACGCGGGCATAAAC |
| AGBE Y314S FP | CTGCGTTTCTTGCTATCCAACCTGCGTTG |
| AGBE Y314S RP | CACCTCGTACTCCACTGAGTTGAAGAGACGACTGTC |
| attB1 hGBE1 FP | TCAAGTTTGTACAAAAAAGCAGGCTATGGCGGCTCCGATGACTC |
| attB2 hGBE1 RP | CCCACTTTGTACAAGAAAGCTGGGTTCAATTCGGCAGATCCACATTC |
| attB1 hIRP1 FP | CAAGTTTGTACAAAAAAGCAGGCTATGAGCAACCCATTCGCAC |
| attB2 hIRP1 RP | CCCACTTTGTACAAGAAAGCTGGGTCTACTTGGCCATCTTGCGGATC |
| attB1 Cisd2 FP | ACAAGTTTGTACAAAAAAGCAGGCTATGGAGCCCATATCACATCTG |
| attB2 Cisd2 RP | ACCACTTTGTACAAGAAAGCTGGGTTCTTGATGACAATTGGTC |
| pAFW attB1 RP | AGCCTGCTTTTTTGTACAAACTTGATACCGGTGCTTGTCATCGTCATC |
| pAFW attB2 FP | ACCCAGCTTTCTTGTACAAAGTGGGACGTAAGCTAGCAGGATCTTTG |
| pAMW attB1 RP | AGCCTGCTTTTTTGTACAAACTTGATACCGGTGATTCAAGTCCTCTTC |
| pAMW attB2 FP | ACCCAGCTTTCTTGTACAAAGTGGGACGTAAGCTAGCAGGATCTTTGTG |
| pAHW attB1 RP | AGCCTGCTTTTTTGTACAAACTTGATACCGGTGTCCGCCATGAGCAG |
| pAHW attB2 FP | ACCCAGCTTTCTTGTACAAAGTGGGACGTAAGCTAGCAGGATCTTTG |
| pAc5 STABLE2 RP | CATGGTGGCGAATTCCACCAC |
| pAc5 STABLE2 FP | GAGGAAGTCTTCTAACATGCGGTGACGTGGAGGAGAATCCCGGCCCT |
| pAc5 3xFlag FP | TGGTGGAATTCGCCACCATGGACTACAAAGACCATGACGGTG |
| pAc5 3xFlag T2A RP | ACCAGGGCCAGGGTTCTCTTCGACATCTCCGCAAGTCAGTAGGCTGCCGCGTCCTTCGCGGCCCCACTTTGTACAAGAAAG |
| pAc5 3xFlag T2A 6xMyc FP | TACTGACTTGCGGAGATGTCGAAGAGAACCCTGGCCCTGGTTCCGATATCTCTAGAGCCACCGAGCAAAAGCTCATTTCTGAAG |
| pAc5 6xMyc attB2 RP | CGGGATTCTCCTCCACGTCACCGCATGTTAGAAGACTTCCTCTGCCCTCAAGCCACTTTGTACAAGAAAG |
| pAc5 6xMyc T2A RP | CGGGATTCTCCTCCACGTCACCGCATGTTAGAAGACTTCCTCTGCCCTCAAGCCACTTTGTACAAGAAAG |
| pAc5 3xHA FP | AGGAAGTCTTCTAACATGCGGTGACGTGGAGGAGAATCCCGGCCCTGCTAGCTACCCATACGATGTTCCTGAC |
| pAc5 3xHA attB2 RP | TCATGTCTGGATCCCTCGAGCCCACTTTGTACAAGAAAGCTG |
| Real-time PCR | |
| SdhB FP | ACGAGCAGTACCGCAACAT |
| SdhB RP | GGCCTTGCCCTCTTCTC |
| AGBE qPCR FP | GGCCGTTTGAGCATGAGA |
| AGBE qPCR RP | CGCTTTGGTTTATCTTATTCAGC |
| rp49 qPCR FP | TTCCTTGACGTGCCAAAACT |
| rp49 qPCR RP | AATGATCTATAACAAAATCCCCTGA |

A list of primers and their sequences that were used for generating S2 cell constructs, transgenic and CRISPR constructs, PCR-based mutagenesis, as well as for quantitative real-time PCR.

(PBS3T) for 3× 10 min. Samples were blocked at RT for 1 h in blocking solution (1× PBS3T 5% normal goat serum (Abcam #ab138478)) and incubated in primary antibody dilution buffer (antibody diluted in 1× PBS3T and 1% BSA) overnight at 4 °C with gentle shaking. Samples were then washed in 1× PBS3T for three times with 10 min each, incubated in secondary antibody dilution buffer for 1 h at RT, washed in 1× PBS3T and 1:50,000 DAPI (Cell Signaling #4083) for three times. Samples were mounted in Vectashield mounting medium (#VECTH1000). Pictures were taken on Nikon Eclipse 80i Confocal C2 + microscope/camera. We used the following antibodies: a monoclonal mouse anti-Flag-tag antibody (Cell Signaling #8146S), a rabbit monoclonal anti-Flag tag antibody (Cell Signaling #14793S), a monoclonal rabbit anti-Myc-tag antibody (Cell Signaling #227S). Primary antibody sera were used at a ratio of 1:400 for endogenously tagged proteins and 1:1000 for

transgenically expressed proteins. Secondary antibodies were obtained from Abcam and used at 1:2000 ratio, including goat anti-rabbit IgG H&L (Alexa Fluor 488) (#ab150077), Alexa Fluor 555 (#ab150078), goat anti-mouse IgG H&L Alexa Fluor 488 (#ab150113), and Alexa Fluor 555 (#ab150114).

**Ferric iron staining**. This protocol was modified from Perl's staining for iron to reduce background noise, a common issue with iron-staining techniques. 42-h L3 larvae were washed in 1× PBS for three times and dissected for BRGC. Samples were fixed with 1× PBS/4% formaldehyde for 20 min at RT. BRGC were washed 1× 10 min, 1× 20 min and 1 × 30 min in 1× PBS/0.3% Triton. Samples were incubated at RT for 1 h in fresh staining solution (2% $K_4Fe(CN)_6$ + 2% HCl) and

briefly washed in 1× PBS/0.3% Triton for 5 × 2 min. Samples were then incubated in 0.01 NaN₃/0.3% H₂O₂ for 30 min at RT and washed 3× 10 min in 0.1 M Phosphate buffer pH 7.0 (57.75 mM $Na_2HPO_4$ and 42.25 mM $NaH_2PO_4$). Samples were then incubated for 10 min with fresh intensification buffer (0.1 M phosphate buffer pH 7.0 containing 0.00125% DAB and 0.0025% $CoCl_2$) to reduce background staining, followed by 3× 10 min wash steps in 0.1 M phosphate buffer pH 7.0. Images were taken using epifluorescence camera (Nikon Digital Sight DS-U3).

**RNA-sequencing.** Animals were reared on standard NutriFly media (Diamed). For a single biological replicate, 50 ring glands were manually dissected in 1× PBS, transferred to Trizol (ThermoFisher #15596026), and flash-frozen in liquid nitrogen for long-term storage. RNA was extracted with the RNAeasy kit (Qiagen #74106) coupled to an on-column DNA digestion step using RNA-free DNAse (Qiagen #79254). Extracted RNA was examined on a Bioanalyser using Agilent RNA 600 nano kit (#5067-1511) to confirm RNA integrity. 100 ng total RNA from each sample was used for generating strand-specific RNA-Seq libraries based on the Ovation *Drosophila* RNA-Seq System 1-16 (Nugen #0350-32). cDNA quality was analysed on a Bioanalyser using the high sensitivity DNA analysis kit (Agilent #5067-4626). 100 ng cDNA in 25 μl nuclease-free water was used for RNA-Seq analysis (Genome Quebec Innovation Center at McGill University). Sequencing data was analysed using Arraystar 4.0 (DNAstar), MS Access and DAVID GO Tools[58]. All RNA-Seq data has been deposited with GEO (entry # GSE130103).

**Ex vivo culturing of ring glands.** In the first approach, BRGC were isolated from $w^{1118}$ L3 larvae just after the L2/L3 moult, transferred to culture medium (Schneider insect medium with 10% heat-inactivated FBS, 1% streptomycin-penicillin, 10 μg/ml insulin and 2 μg/ml ecdysone), and incubated at 25 °C. These conditions efficiently mimicked in vivo conditions and allowed physiological functions to be studied for up to 48 h[59]. To reduce available iron, BPS was added to the culture medium at a final concentration of 100 nM. After 24 h, ring glands (50 per replicate) were transferred to Trizol for later qPCR analysis. In a separate approach, S2 cells were maintained under the same conditions, with a starting titer of $1 × 10^6$ cells per ml. After 24 h at 25 °C, 3 ml were used for RNA extraction and qPCR analysis. For the in vivo approach, $w^{1118}$ larvae were initially reared on regular NutriFly food, and after the L2/L3 moult transferred to NutriFly food containing 100 μM BPS. After 24 h, ring glands were dissected and subjected to RNA extraction and qPCR analysis. For primers see Table 3.

**Quantitative real-time PCR (qPCR).** Extracted RNA (Qiagen RNeasy extraction kit) was reverse-transcribed via ABI High Capacity cDNA synthesis kit (ThermoFisher #4368814). Synthesized cDNA was used for qPCR (QuantStudio 6 Flex) using KAPA SYBR Fast qPCR master mix #Sigma KK4601). For each condition, three biological samples were each tested in triplicate. Samples were normalized to *rp49* based on the ΔΔCT method, with the exception of RNA-immunoprecipitation, where we normalized results to immunoprecipitated IRP1A protein levels. For primers see Table 3.

**Constructs for co-immunoprecipitation (S2 cells).** Fragments carrying *Drosophila* AGBE and human GBE1 cDNAs were cloned into pAMW while *Drosophila* IRP1A and human IRP1 cDNAs were cloned into pAFW. The *Drosophila* Cisd2 cDNA was cloned into pAHW, and eGFP was cloned into pAFW as well as pAMW. This approach allowed for the generation of in-frame tagged cDNAs. We used an approach that allows for the co-expression of two cDNAs that are separated by a viral-derived 2A-like peptide, which is then cleaved post-translationally to yield equal amounts of both proteins[60]. Appropriate pairwise combinations of cDNAs encoding wild type or modified versions of 6x Myc-tagged *AGBE* cDNA, 6x Myc-tagged human *GBE1*, 3x Flag-tagged *IRP1A*, 3x HA-tagged Cisd2 and 3x Flag-tagged eGFP (the latter served as a control) were cloned into pAc5-STABLE2-Neo (Addgene #32426). For the triple transfection of *IRP1A*, *AGBE* and *Cisd2*, 3x Flag-tagged *IRP1A* was cloned together with 6x Myc-tagged *AGBE* as well as 3x HA-tagged *Cisd2* into pAc5-STABLE2-Neo, separated by viral-derived 2A-like peptides.

**Transfection, co-immunoprecipitation and Western Blotting.** Cells were grown in Schneider Insect medium with 10% heat-inactivated FBS, 1% Streptomycin-Penicillin following standard procedures and transfected by the Calcium Phosphate-based method (Invitrogen). Transfected cells were lysed, and Myc-tagged bait proteins were immunoprecipitated using Myc-trap agarose beads (Chromotek Myc-Trap®-A). Flag-tagged bait proteins were immunoprecipitated using M2 Flag agarose beads (Sigma-Aldrich #A2220) following instructions of the manufacturer. Pulled-down samples were analysed via Western Blotting. To detect 3x Myc-tagged proteins, monoclonal rabbit anti-Myc-tag antibodies (Cell Signaling #2278S) were used at a concentration of 1:2,500 followed by incubation with a goat anti-rabbit IgG H&L HRP secondary antibody (Abcam #ab97051) at a ratio of 1:10,000. To detect 3x Flag-tagged proteins, monoclonal mouse anti-Flag-tagged antibodies (Cell Signaling #8146S) were used at a concentration of 1:2,500 followed by incubation with Goat anti-mouse IgG H&L HRP secondary antibodies (Abcam #97023) at a ratio of 1:10,000. 3x HA-tagged proteins were

detected using either monoclonal mouse anti-HA-tag antibodies (Abcam #18181) or monoclonal rabbit anti-HA-tag antibodies (Cell Signaling #3724S) both at a ratio of 1:2,500 followed by incubation with either Goat anti-mouse or Goat anti-rabbit IgG H&L HRP as a secondary antibody, respectively, at a ratio of 1:10,000. Blots were scanned for image acquisition with a ChemiDoc imaging system (Bio-Rad). Uncropped scans of all Western Blots are provided in Supplementary Figs. 12–13.

**Mass spectrometry of whole larvae.** Our whole-body mass spectrometry (MS) approach was adapted from an in vivo cross-linking procedure developed for *Drosophila* embryos[61,62]. We collected 150–200 L3 larvae (40–42 h after the moult) and washed them 3 × 5 min in 1× PBS. Animals were then incubated in 1× PBS with 0.1% Triton (PBS1T) 2 × 5 min before fixing in fresh fixative solution (1× PBS1T with 0.2% Formaldehyde) for 10 min. The fixing solution was removed and replaced by fresh quenching solution (0.25 M glycine in 1× PBS1T). Animals were washed in 1× PBS1T three times before being flash-frozen in liquid nitrogen for long term storage at −80 °C. Larvae were homogenized in 1 ml of 1× lysis buffer (25 mM Na-HEPES pH 7.5, 75 mM NaCl, 0.5 mM EDTA, 10% glycerol, 0.1% Triton X-100, proteinase inhibitor cocktail (Sigma #11873580001)) using a Dounce homogenizer. Lysates were centrifuged at 16,000 g for 30 min at 4 °C. Protein concentrations of supernatants were determined with the QubitTM Protein assay (Invitrogen #Q33212) and served to equalize protein amounts for subsequent co-immunoprecipitation assays. The supernatants were then transferred to spin columns (Chromotek sct-50) and incubated with 40 μl of anti-Flag M2 affinity gel (Sigma #A2220) on a rotating shaker for 2 h at 4 °C. Columns were centrifuged and treated with wash buffer 1 (25 mM Na-HEPES pH 7.5, 75 mM NaCl, 0.5 mM EDTA, 10% Glycerol, 0.1% Triton X-100) and wash buffer 2 (25 mM Na-HEPES pH 7.5, 75 mM NaCl, 0.5 mM EDTA, 10% Glycerol) for three times each. At the last step, 40 μl of loading buffer (0.125 M Tris-HCl pH 6.8, 5% SDS, 0.004% Bromophenol blue, 20% glycerol, 1.43 M β-mercaptoethanol) was added and tubes were incubated at 95 °C for 5 min before collecting samples. Samples were then loaded on a 12.5% SDS-gel, stained with Coomassie Blue and submitted for MALDI-TOF MS analysis (carried out by the Alberta Proteomics and MS Facility, University of Alberta).

In brief, we performed overnight in-gel trypsin digestion following standard procedures. Gel bands were excised and destained twice in 100 mM ammonium bicarbonate (Sigma #09830-500 g)/acetonitrile (Sigma #271004) at a ratio of 50:50 (v/v). Samples were then reduced using 10 mM β-mercaptoethanol (Sigma #M6250) in 100 mM bicarbonate, followed by alkylation in 55 mM iodoacetamide (Sigma #I11490) in 100 mM bicarbonate. After dehydration, a trypsin solution (Promega #V5111) was added to cover the gel pieces at a final concentration of 6 ng/μl, and digested overnight (~16 h) at RT. Tryptic peptides were first extracted from the gel using 97% water/2% acetonitrile/1% formic acid followed by a second extraction using 50% of the first buffer and 50% acetonitrile.

Fractions containing tryptic peptides were resolved and ionized by using nanoflow HPLC (Easy-nLC II, Thermo Scientific) coupled to an LTQ XL-Orbitrap hybrid mass spectrometer (Thermo Scientific). Nanoflow chromatography and electrospray ionization were carried out with a PicoFrit-fused silica capillary column (ProteoPepII, C18) with a 100 μm inner diameter (300 Å, 5 μm, New Objective). Peptides were loaded onto the column at a flow rate of 3000 nl/min and resolved at 500 nl/min using a 60 min linear gradient from 0 to 45% v/v aqueous acetonitrile in 0.2% v/v formic acid. The mass spectrometer was operated in data-dependent acquisition mode, recording high-accuracy and high-resolution survey Orbitrap spectra using external mass calibration, with a resolution of 30,000 and m/z range of 400-2000. The fourteen most intense multiply charged ions were sequentially fragmented by using collision-induced dissociation, and the spectra of their fragments were recorded in the linear ion trap. After two fragmentations, all precursors selected for dissociation were dynamically excluded for 60 s.

Data were processed using the Proteome Discoverer 1.4 software (Thermo Scientific), and we used the SEQUEST search algorithm (Thermo Scientific) to identify peptides in the UniProt (uniprot.org) *Drosophila* proteome database (ID: UP000000803). For this approach, search parameters included a precursor mass tolerance of 10 ppm and a fragment mass tolerance of 0.8 Da. Peptides contained carbamidomethyl cysteines as static modifications and oxidized methionines and deamidated glutamines and asparagines as dynamic modifications. Minimum acceptance criteria were based on two or more peptides per detected protein, and up to two missed cleavage protease cleavage sites (C terminal sites of lysine and arginine residues) were tolerated for peptide matching. Peptide-to-spectrum matches (PSM) were set to 1. We used the target decoy approach[63,64] to estimate false discovery rates (FDRs) for peptide matches, with the strict FDR value equalling 0.01 and a relaxed FDR of 0.05. For our analysis, the signal to noise threshold was set to 1.5. The SEQUEST protein raw score was used to calculate the final mass spectrometry score for each detected protein. The raw score represents the sum of (i) matched peptide fragment ion intensities, (ii) the number of total and matched fragment ions, and (iii) the factor that reflects the continuity of a matching ion peptide series for a given protein. SEQUEST then determines the final scores by converting the expected masses of peptide ions into a theoretical spectrum, combined with a calculation that determines the cross-correlation between the theoretical spectrum and the experimental spectrum[65]. In total, we analysed 21 samples representing 17 different conditions, four of which were tested

twice (Supplementary Table 2). The reproducibility of the repeated biological samples ranged from 82.4–98.4% (Supplementary Table 2), while the overlap between equivalent IRP1A and IRP1B variants (e.g., IRP1A$^{3R3Q}$ vs. IRP1B$^{3R3Q}$) ranged from 63.8% to 66.3%. In total, five control samples were used (Supplementary Table 2). Proteins detected in any of the control samples were removed from experimental results.

**Mass spectrometry of the prothoracic gland (PG)**. For PG-specific MS we separated hand-dissected BRGC into individual cells. We used larvae that expressed Venus-tagged CD8 (UAS-CD8.Venus, Bloomington stock #65609) in a PG-specific manner. CD8 localises to the cell membrane, allowing purification of PG cells from unlabelled cells[66]. BRGC were collected in ex vivo medium (Schneider insect medium with 10% heat-inactivated FBS, 1% streptomycin-penicillin, 10 µg/ml insulin and 2 µg/ml ecdysone) containing a proteinase inhibitor cocktail (Sigma #11873580001). Dissection times were limited to 1 h to minimize physiological changes. Samples were incubated in 1× PBS1T for 2 × 5 min before being fixed in fresh fixing solution (1× PBS1T containing 0.2% formaldehyde) for 10 min. Fixing solution was removed and replaced by fresh quenching solution (0.25 M glycine in 1× PBS1T). Samples were washed three times in 1× PBS1T, followed by immersion in 1× PBS1T/25% glycerol and flash-frozen in liquid nitrogen for long-term storage at −80 °C. For knock-in derived proteins we collected 1.0 ml containing the equivalent of ~15,000 BRGCs isolated from 40–42 h old L3, while roughly half the amount was used for flies with transgenically produced protein. Samples were removed from −80 °C and thawed gradually for 15 min at −20 °C followed by 15 min at 4 °C until completely thawed. Tissue samples were then incubated in cell dissociation buffer (CMF buffer with 1 mg/ml collagenase, 1 mg/ml papain) for 30 min at 30 °C. The digestion was terminated by adding 4x volumes of CMF to the dissociation reaction. Samples were left at RT for 5 min before being centrifuged at 1000 g for 1 min. Cells were 3× washed in PBS1T and incubated with IgG beads that had been cross-linked with mouse CD8 antibody (#ab82005) for 30 min, followed by three brief washes in PBS1T, and an elution step (0.1 M citrate pH 2.3) to release PG cells from beads. All subsequent steps for protein extraction and immunoprecipitation were as described for whole-body MS. All MS proteomics data have been deposited to the ProteomeXchange Consortium[67] via the PRIDE[68] partner repository with identifier #PXD013499.

**Quantitative RNA-immunoprecipitation (RIP)**. Our in vivo RIP approach was adapted from different cell culture protocols[69–71]. As controls, we used $w^{1118}$, which is the parental strain for our transgenic and mutant lines and thus harbours no tagged genes. To immunoprecipitate IRP1A, we used 3x Flag-tagged CRISPR/Cas9-generated knock-in alleles, namely IRP1A$^{3F}$, IRP1A$^{C450S.3F}$, and IRP1A$^{3R3Q.3F}$, representing tagged wild type, constitutively RNA-binding and non-RNA-binding forms of IRP1A, respectively. We collected 200 L3 larvae (staged at 40 h after the L2/L3 moult) per sample. Larvae were washed for 3 × 5 min in PBS, flash-frozen in liquid nitrogen and stored at −80 °C. Larvae were homogenized in 1 ml lysis buffer (150 mM KCl, 25 mM Tris-HCl pH 7.4, 5 mM EDTA, 0.5% v/v Nonidet P-40, 1× proteinase inhibitor cocktail, 100 U/ml RNase inhibitor (NEB #M0314S) using a Dounce homogenizer. Lysates were centrifuged at 12,000 g for 30 min at 4 °C. Supernatants were transferred and filtered through a 0.45 µm syringe filter (Sigma #CLS431225-50EA). Flow-through samples were incubated with 300 µl equilibrated anti-Flag M2 affinity gel solution on a rotating platform for 4 h at 4 °C followed by centrifugation at 12,000 g for 30 min at 4 °C. The supernatant was removed, and the affinity gel washed in 10x volume of lysis buffer for 2 × 5 min. 5% of the final volume was saved for Western Blotting to determine IRP1A levels to adjust sample input for RIP (Fig. 2c). Western Blots were scanned with the ChemiDoc imaging system (Bio-Rad), and bands were quantified using ImageJ following standard procedures. The remaining 95% was used for Trizol-based RNA extraction followed by qPCR for SdhB, which harbours a validated IRE[72,73]. For primers see Table 3.

**Measuring IRP1A and IRP1B aconitase activity (S2 cells)**. S2 cells were grown in Schneider Insect medium with 10% heat-inactivated FBS/1% Streptomycin-Penicillin and transfected by the Calcium Phosphate-based method (Invitrogen). Transfected cells were lysed, and IRP1 protein levels were evaluated as follows: From each sample, 50% of the lysate was used to immunoprecipitate IRP1A or IRP1B, and proteins were separated via SDS-PAGE. This was followed by Coomassie Blue staining of the gel to evaluate IRP1A and IRP1B protein levels, and cell lysate amounts used for aconitase assays were normalized accordingly. Aconitase activity was determined by measuring the rate of NADPH production via absorbance at 340 nm every 5 min (Aconitase-340$^{TM}$ kit, OxisResearch 21041, DU-730 UV/Vis Spectrophotometer). The absorbance rate was normalized relative to the rate of untransfected S2 cells, which served as a negative control for background aconitase activity.

**Measuring IRP1A and IRP1B aconitase activity in vivo**. We measured aconitase activity from both transgenically produced IRP1 (Supplementary Fig. 7) as well as from knock-in alleles (Fig. 2d). For the former, we collected 200–250 L1 larvae that ubiquitously expressed transgenic IRP1A or IRP1B alleles (tub-Gal4 > UAS-cDNA) and washed them 3 × 5 min in 1 × PBS. To measure the IRP1 aconitase activity in an AGBE mutant background, we generated lines carrying either

transgenic tub > IRP1A or tub > IRP1B together with transgenic UAS-FLP-cDNA and the AGBE$^{FCF}$ knock-in allele to remove AGBE ubiquitously. For corresponding controls, we used the same combination, except that we replaced UAS-FLP-cDNA with a UAS > eGFP-cDNA transgene. To evaluate the aconitase activity produced by knocked-in IRP1A alleles, we collected 200 L3 larvae (staged at 42 h after the L2/L3 moult) carrying different IRP1A alleles in an IRP1B null mutant background to eliminate IRP1B aconitase activity. Larvae were homogenized in 1 ml of 1× Lysis buffer (25 mM Na-HEPES pH 7.5, 75 mM NaCl, 0.5 mM EDTA, 10% glycerol, 0.1% Triton X-100, proteinase inhibitor cocktail). To reduce the contribution of mitochondrial aconitase (Acon), we removed the mitochondrial fraction via ultracentrifugation at 20,000 g and 4 °C. We normalized samples based on immunoprecipitated tagged protein levels (i.e., IRP1A variants), which we evaluated via Western Blotting (Fig. 2d, not shown in Supplementary Fig. 7), as described in the quantitative RIP section above. Aconitase activity was determined as described for S2 cells. For the knocked-in CRISPR alleles of IRP1A, we used IRP1B null mutants as controls, which harbour a wild type copy of IRP1A.

**Reporting summary**. Further information on research design is available in the Nature Research Reporting Summary linked to this article.

## Data availability
The source data underlying Figs. 1g–h, 2a–d, 3a, 3c. 4e–g, 4i and Supplementary Fig. 7 are provided as a Source Data file at FigShare (https://figshare.com/s/eb4451ceeae5e0fd926f) with DOI information https://doi.org/10.6084/m9.figshare.8001809. Mass spectrometry proteomics data relating to Fig. 4a, b and Supplemental Tables 3–6 have been deposited with the ProteomeXchange Consortium via the PRIDE partner repository (identifier #PXD013499). RNA-Seq data have been deposited with GEO (https://www.ncbi.nlm.nih.gov/geo/query/acc.cgi?acc = GSE130103).

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

## Acknowledgements

The authors thank the Bloomington *Drosophila* Stock Center at Indiana University, and the Vienna *Drosophila* Resource Center for sending fly stocks. We thank the labs of Henry Krause and Michael O'Connor for providing fly stocks, and we also thank Andrew Simmonds for sharing vectors and cell culture tools with us. We extend our thanks to Fanis Missirlis and Arash Bashirullah for insightful comments and discussions. K.K.J. wishes to thank the CIHR for supporting this work (MOP 93761). R.L. acknowledges the generous financial support from Deutsche Forschungsgemeinschaft (Koselleck grant LI 415/6), and networking support from the COST Action FeSBioNet (Contract CA15133).

## Author contributions

N.H. co-designed experiments and carried out most experiments, Q.O. conducted the RNAi screen and initial phenotypic characterization, P.C. generated IRP1 transgenic lines and analyzed NOS[IR-X] phenotypes. R.L. trained Q.O. in iron-related experiments, provided reagents and had an advisory role throughout the project. K.K.J. analyzed the RNA-Seq data, supervised and co-designed experiments, and wrote the manuscript with input from the other authors.

## Competing interests

The authors declare no competing interests.
