## [Peer Review File · Nature Communications]

Reviewers' comments:

Reviewer #1 (Remarks to the Author):

The manuscript by Huynh provides unanticipated, novel and potentially far-reaching findings regarding the regulation of iron homeostasis in metazoans. Using a series of genetic and biochemical studies focused on the *Drosophila* prothoracic gland they demonstrate several highly impactful findings: 1. The glycogen branching enzyme AGBE has a critical and completely unanticipated role in iron metabolism as noted by the development of iron-dependent porphyria-like symptoms in the PG in AGBE deficient embryos. 2. The finding that IRP1A but not an RNA binding only form of IRP1A can rescue AGBE deficient embryos suggesting a role for the aconitase function of IRP1 in the proper control of iron metabolism. 3. That IRP1A is essential for larval survival under iron-deficient conditions and IRP1A deficiency phenocopies the porphyria phenotype of AGBE mutants. 4. That IRP1A and IRP1B accumulate in the nucleus in an iron and Fe-S cluster dependent manner and that the nuclear accumulation of IRP1A requires AGBE. 5. That deficiency of Cisd2 (MitoNEET), a protein that can function in the repair of the Fe-S cluster in IRP1, phenocopies the IRP1A and AGBE mutants. 6. Evidence suggesting that nuclear IRP1 may be involved in transcriptional regulation of genes involved in iron metabolism and iron-requiring processes. While the manuscript contains data with paradigm shifting implications, particularly with respect to the role of AGBE in iron metabolism and its interaction with IRP1, there is concern with regard to some of the findings and approaches that need to be addressed. Overall, the work described here has the potential to have a major impact on the concepts that describe our understanding of the control of iron metabolism in metazoans.

1. The authors make use of a number of IRP1 mutants to assist in the analysis of the combined role of IRP1 with AGBE. In particular, they use a 3R3Q mutant focusing on R536,541 and 793. As the authors note Philpott et al mutated these arginines to glutamine and found significant reductions in RNA binding. However, Philpott only analyzed single mutants and did not make the triple mutant used here. Furthermore, as noted by Philpott, these 3 residues in aconitases are in the enzyme active site and are essential for substrate binding and catalysis. On this basis it seems likely that the 3R3Q mutant used here lacks both RNA binding and aconitase functions. Thus, the statement on line 162 (see also line 299) that the 3R3Q mutant is an aconitase only form requires for direct demonstration that this is the case.

Furthermore, given the simultaneous introduction of 3 mutations, along with the proposal that the 3R3Q mutant represents holo-IRP1 (with FeS cluster), raises the question as to whether this mutant can efficiently (or at all) incorporate an Fe-S cluster in cells and whether or not it can fold properly. Use of this mutant provides a key basis for several findings but it seems likely that it lacks both RNA binding and aconitase activities and the extent to which it can incorporate an Fe-S cluster is not clear which all together clouds the interpretation of some of the results. In this regard, the inability of the 3R3Q mutant to rescue the loss-of-AGBE-function phenotypes (line 165) and the suggestion that the aconitase function is not sufficient for rescue should be reconsidered.

In addition, in some cases the authors appear to use a 3R3K mutant (line 143 and figure 2B) but in most other cases its 3R3Q. Is this correct? If so, please explain. If not, then the authors need to demonstrate that the 3R3Q mutant interacts with AGBE by co-IP.

Also, on line 143 the authors refer to a A793K mutant. Is 793 not an R?

2. The data in figure 2B shows co-IPs of IRP1 wildtype and mutant IRPs with AGBE. Most notably they show that the C450S mutant which should not be able to bind an Fe-S cluster, fails to interact with AGBE. A blot showing the amount of input for IRP1 variants and also AGBE is needed. Mutants of IRP1 that cannot bind a FeS cluster are unstable and do not accumulate as much as the wildtype IRP1. Is the lack of interaction of the C450S mutant of IRP1 shown in Figure 2B affected

by the steady state level of this variant?

More direct evidence is needed that it is the holo-form of IRP1 that binds to AGBE.

In this regard, are the co-IPs quantitative or is some of the (over?)expressed IRP1 mutants not brought down by the AGBE immunoprecipitation? If this is not quantitative then the extent to which these mutants bind a cluster and whether cluster-containing vs. cluster-devoid species of the mutants is present could affect the results and interpretation.

3. The authors provide intriguing data that IRP1 may drive regulation of transcription of genes involved in iron dependent processes. The data in table 2 compares IRP1A vs IRP1A(3R3Q) and notes the much stronger impact of the mutant. Since the 3R3Q mutant used likely lacks both RNA binding and aconitase functions the interpretation of these interesting findings requires reevaluation as does the model in figure 5 which suggests a role for the aconitase function of IRP1 in the nucleus.

4. The authors provide interesting suggestions about the possible tissue specific functions of IRP1 (i.e. line 215). Some comments are needed about the tissue specific expression of AGBE in flies and other species.

5. Protein-protein interaction analysis by MS provides important support for the genetic interactions between AGBE, IRP1 and mitoNEET. The authors comment that IRP1 is found to interact with translation factors and also with 40 ribosomal proteins. The latter seems surprising, are the ribosomal protein interactions all direct? If not, or if this is not examined, it should be noted.

The findings that the C450S mutant of IRP1, which cannot be an aconitase, does not interact with AGBE supports the view that the aconitase form is the relevant species. However, without demonstration that it is functional in some way (ie. it can load a FeS cluster), the use of the 3R3Q mutant does not address this important issue.

6. Several comments are made in the manuscript about the presumed complete conversion of the aconitase form of IRP1 to the RNA binding form in iron deficiency. The Fe-S cluster of cytosolic aconitase is rather stable and the impact of iron deficiency does not necessarily lead to complete conversion to the RNA binding form

Reviewer #2 (Remarks to the Author):

Authors have identified an important molecular mechanism that regulates iron concentration and haem-containing proteins in the cell. It is fascinating that a protein known for more than a decade for its function in branching glycogen chains is involved in iron regulation. Although recent findings would give us a hint about this mechanism, no evidence has been provided (articles related to RBCK1 and polyglyglucosan body myopathy). I was surprised authors didn't pick this and mentioned in the discussion. GBE has been studied in glycogen metabolism for many years it has been shown that its act together with the activity of glycogen synthase ensures proper branching of glycogen. A slight disturbance in this delicate balance (towards Glycogen synthase) causes longer glucose chains. Higher activity of glycogen synthase changes the structure of glycogen to starch (Adult polyglucosan body disease APBD, seldom branches) or amylose (Andersen's disease no branches). Polyglucosan (PG) is the apparent product of GBE deficiency. Therefore, it is the hallmark of Glycogenesis Type IV (GSDIV). However, PG also forms in other genetic diseases. Genes that are unrelated to glycogen synthesis or metabolism such as Lafora disease and RBCK1 deficiency. Lafora disease is caused by the deficiency of laforin or Malin (E3 ubiquitin ligase), the other PGB disease is caused by the deficiency of RBCK1 or HOIL1 (Heme-Oxidized IRP2 Ubiquitin

Ligase 1). It is still a debate if the PG or energy deficiency is the leading cause of the disease. It is exciting and a great addition to the knowledge in glycogen metabolism and the proteins involved in it, this article brings a third mechanism for the pathogenesis of GSD IV.

Authors incorporated all alternative mechanisms and controls. I congratulate for their effort.

Besides the addition to the discussion above, I would like to correct a statement made on line 106 "final steps of glycogen synthesis" glycogen synthesis is composed of three steps 1-formation of the primer by glycogenin dimers, 2-extension of glucose chains by glycogen synthase 3- formation of branches by GBE and further extension of branches by glycogen synthase. It will be better to remove the "final steps."

All the protein interactions are shown by immune precipitation and MS. There is no immunostaining and confocal microscopy to back up the data for colocalization of proteins to mitochondria. Proteins may interact with each other in homogenized samples that might be misleading. I suggest authors show at least mitochondrial localization of AGBE. This is not shown before. It is also essential to show the function and protein-protein interactions in case of deleterious mutations in AGBE, such as p.Y329S substitution, tyrosine is a conserved amino acid in both vertebrate and Drosophila homologs. This is particularly important because the p.Y329S mutation is in a linker region between amylase and translocase domains, where there is no enzymatic activity is involved. However, it is the most common underlying genetic cause of APBD. Also, there is no information about which part (domain) of the AGBE mediates physical interaction with other proteins.

Reviewer #3 (Remarks to the Author):

This is a very nice and extensive study that begins with a relatively novel screen for knockdowns that cause overgrowth and altered protoporphyrin metabolism in the Drosophila prothoracic gland, and then goes on to make a number of unappreciated and unexpected links between new regulators of iron metabolism. Given the broad general importance of the processes analyzed, and the extensive use of complementary genetic and molecular approaches, it seems highly deserving of publication in Nature communications. There were however some apparent mixups, areas of confusion and omissions that need to be dealt with. I will go through these as they arise in the manuscript.

line 103: I'm confused by "We were unsuccessful in finding independent evidence for the NosIR-X and spz5IR". Could you clarify/restate what you saw with these knockdowns and how they differed? Nos, at least, causes the somehow related PG enlargement, but I gather the red colour is due to heme accumulation as opposed to upstream protoporphyrin accumulation or lack of production. On a related point, if the authors have any idea what the link is between PG size and iron metabolism, if any, it would be helpful for the reader.

line 104: "however, a second, non-overlapping RNAi line targeting AGBE, AGBEIR2, caused similar phenotypes (Figure 1F)." I assume this means similar to ALAS1 KD - enlarged PG without increased fluorescence (no protoporphyrin)?

The authors should do in situ or RT-PCR to confirm that AGBE and IRP1 are expressed and when in the PG (unless already published somewhere). So far, they have only looked at Gal4/UAS induced tagged versions of IRP1A/B. Do these levels change in AGBE or other mutant backgrounds? with Fe depletion/supplementation?

line 128: "However, ubiquitous expression of RNAi caused widespread larval lethality, confirming that all RNAi lines were functional".... Presumably this is referring to AGBE RNAi lines.

134: "A search of protein-protein interaction databases¹⁸ revealed that human GBE1 physically interacts with IRP1" How was this original interaction determined? Was the interaction published and discussed previously, or just annotated?

The following two quotes from the manuscript are contradictory.

138: Only IRP1A has been shown to switch from holo- to the IRE-binding apoform, while IRP1B is believed to act only as an aconitase .

148: "This was paradoxical, as AGBE mutations caused iron-deficiency phenotypes, but holo-IRP1A has no known roles in iron homeostasis as it is believed to only act as an aconitase."
The finding is not paradoxical if line 138 is correct.

In figure 1, why wasn't IRP1B tested for interaction with AGBE? Some redundancy here could explain the weaker IRP1A phenotype as compared to AGBE (discussed further below).

157: "we observed dramatic rescue of PG>NOSIR-X RNAi animals with respect to both the lethality (Figure 2D) and protoporphyrin accumulation (not shown).

Whats the presumed mechanism/explanation for this? Where does NOS fit in? Protoporphyrin or heme accumulation?

line 175: Since both lines are available, I would like to see double IRP1A/B RNAi - I suspect there's some redundancy happening, and that this would result in a stronger phenotype.

line 192: "Taken together, these results showed that only IRP1A has critical roles in iron homeostasis and that IRP1A depletion phenocopied the iron-dependent porphyria seen in AGBE mutants."

If I understand correctly, I disagree. I don't see that a contribution by IRP1B, perhaps lesser and largely redundant in manner, has been ruled out.

IRP1B did not require AGBE for nuclear localization, but this does not mean it cannot act like IRP1A in terms of iron regulation. In fact, mutual pull-downs are consistent with all working together with IRP1A/B overlapping functions. If IRP1B can interact with AGBE and IRP1A, it doesn't need to interact with Cisd2 (MitoNEET) directly in order to provide function. Do IRP1s hetero/homodimerize? If not directly, it sounds like they do indirectly via AGBE or MitoNEET.

There appear to be significant issues with the MS data. The s1 and s2 tables referred to for MS data do not appear to be included. Tables S3 and S4 don't clearly indicate the bait used and where it shows in the recovered peptides/spectra. Proteins generally weren't named, so I could not tell if the proteins mentioned in the results were there in the tables. Legends to explain the tables appear to be missing. The most abundant proteins look like chaperones and other abundant proteins such as ribosomal subunits and histones (actually didn't see the latter). To be real, interactors should be less abundant than the bait, which if well done, should be the most abundant protein in the purification. Anything more abundant is almost certainly background. Comments on data analysis methods (spectral counts, non-specific protein filtering etc) should also be mentioned briefly in the results section.

For future experiments, the authors should consider the possibility that the ability of IRP1s to bind IREs could facilitate interactions with guide RNAs that play a role in nuclear localization and target gene promoter selection. It would be interesting to do RNAseq on affinity purified IRIPs (RIPseq). If RNAs are found, check for IRE's. It should also be considered that many of the interactions detected may be indirect via linking RNA and additional proteins. I think that doing pull-downs in the presence of RNase, and trying direct interactions with purified proteins should be considered (or minimally, the above possibility acknowledged).

Reviewer #4 (Remarks to the Author):

The manuscript by Huynh et al. uncovers two novel findings, using Drosophila as model organism:

(1) AGBE plays a role in maintaining functional IRP1A, likely involving also Cisd2, and (2) IRP proteins translocate to the nucleus, likely in their holo-forms, where they interact with various histone proteins. Finding (1) reveals a previously unknown link between iron homeostasis and glucose metabolism, and finding (2) leads to the suggestion that IRPs in the nucleus might play, via their aconitase activity, a role in regulating the supply of acetyl-CoA for histone acetylation.

The study encompasses a range of transgenic and other modified fly lines, immunostaining, transcriptomic data, and the determination of interactomes for several proteins, for both whole-body and the specific tissue under study (prothoracic gland).

Together, the data support the major findings of the study. The authors have also considered alternative causal relationships, e.g. the possibility that the observed interactions between AGBE and IRPs relates to their (cytosolic) aconitase function.

Although this might be outside the scope of the study, it would have been desirable to have more biochemical information on how the interactions with AGBE might take place, and what the "moonlighting" function of AGBE might be.

In summary, the findings are novel and are likely to influence and stimulate work in this field. A few suggestions for clarifications/improvements are given below.

The interactomes of IRP1A, IRB1B and AGBE were determined by co-immunoprecipitation followed by SDS-PAGE and MALDI-TOF MS. The authors studied a large number of carefully selected samples to address various detailed questions. Interactome data are summarised in Figure 4, and evaluated MS data are presented in a supplementary data file. The Excel data file contains scores that refer to abundances, but it is unclear how these were determined. It would therefore be desirable to provide more detail on sample preparation for MS, and especially the quantitative aspect in these data, on page 21. This should include information on detection limits.

It may also be noted that all "micro" signs seem to have been converted to another symbol.

Line 143: There seems to be an "A" missing in IRP1^{A793K}.

I would like to thank all reviewers for their enthusiastic comments, their constructive criticisms, and for their time spent on going over the material. We believe to have addressed all major concerns, and as a result, the manuscript has improved in some important areas. Please see the original comments in blue italics and our response in regular font beneath.

Reviewer #1 (Remarks to the Author):

The manuscript by Huynh provides unanticipated, novel and potentially far-reaching findings regarding the regulation of iron homeostasis in metazoans.

Thank you!

Using a series of genetic and biochemical studies focused on the Drosophila prothoracic gland they demonstrate several highly impactful findings: 1. The glycogen branching enzyme AGBE has a critical and completely unanticipated role in iron metabolism as noted by the development of iron-dependent porphyria-like symptoms in the PG in AGBE deficient embryos. 2. The finding that IRP1A but not an RNA binding only form of IRP1A can rescue AGBE deficient embryos suggesting a role for the aconitase function of IRP1 in the proper control of iron metabolism. 3. That IRP1A is essential for larval survival under iron-deficient conditions and IRP1A deficiency phenocopies the porphyria phenotype of AGBE mutants. 4. That IRP1A and IRP1B accumulate in the nucleus in an iron and Fe-S cluster dependent manner and that the nuclear accumulation of IRP1A requires AGBE. 5. That deficiency of Cisd2 (MitoNEET), a protein that can function in the repair of the Fe-S cluster in IRP1, phenocopies the IRP1A and AGBE mutants. 6. Evidence suggesting that nuclear IRP1 may be involved in transcriptional regulation of genes involved in iron metabolism and iron-requiring processes. While the manuscript contains data with paradigm shifting implications, particularly with respect to the role of AGBE in iron metabolism and its interaction with IRP1, there is concern with regard to some of the findings and approaches that need to be addressed. Overall, the work described here has the potential to have a major impact on the concepts that describe our understanding of the control of iron metabolism in metazoans.

1. The authors make use of a number of IRP1 mutants to assist in the analysis of the combined role of IRP1 with AGBE. In particular, they use a 3R3Q mutant focusing on R536,541 and 793. As the authors note Philpott et al mutated these arginines to glutamine and found significant reductions in RNA binding. However, Philpott only analyzed single mutants and did not make the triple mutant used here. Furthermore, as noted by Philpott, these 3 residues in aconitases are in the enzyme active site and are essential for substrate binding and catalysis. On this basis it seems likely that the 3R3Q mutant used here lacks both RNA binding and aconitase functions. Thus, the statement on line 162 (see also line 299) that the 3R3Q mutant is an aconitase only form requires for direct demonstration that this is the case.

This is an interesting issue. First off, Philpott et al. never showed that the R->Q point mutations in IRP1 affected Aconitase function, they simply identified these residues based on alignments with human mitochondrial aconitase (where those residues are indeed important for substrate recognition). Secondly, it is important to keep in mind that there is potentially a big difference between “holo-IRP1” and “aconitase”, because we do not know by which mechanisms holo-IRP1 exerts its functions in the nucleus. In any case, to address this directly, we used two approaches. In the first line of experiments, we used cDNA transfection in S2 cells and transgenic fly lines to express wild type IRP1A, IRP1A_C450S and IRP1A_3R3Q in these systems. We then measured aconitase activity in extracts from S2 cells and flies (see new Figure S7). This worked very well, and showed that IRP1A_C450S had no aconitase activity, and that IRP1A_3R3Q aconitase activity was comparable to that of wild type IRP1A (in S2 cells) or slightly less (in transgenic flies). Very similar results were seen for the corresponding IRP1B alleles (Figure S7B + 7D). For S2 cells, we also generated 4R4Q mutant cDNAs for IRP1A and IRP1B, and in both cases this caused strongly reduced aconitase activity.

In addition to the aconitase assays, we also used a complex genetic approach, for which I have attached a figure to the end of this document. We wish to publish the results in a future paper, but we are making them available here for assessment. Basically, we showed by genetic means that the IRP1A_3R3Q allele is functional, because it can rescue animals that only have the apo-form of IRP1A (IRP1A_C450S) as a second allele. In brief, homozygous knock-in alleles for either IRP1A_3R3Q or IRP1A_C450S are 100% lethal on media containing the iron chelator BPS, with no animals reaching adulthood (in a homozygous IRP1B deletion background). This indicates that neither the apoform nor the holoform are sufficient to ensure survival. However, a heterozygous IRP1A_3R3Q / IRP1A_C450S flies are fully viable, even in the absence of IRP1B and on BPS media, a beautiful example of interallelic complementation. This strongly suggests that the switch between holo- and apo-form can be bypassed if one provides one allele capable of RNA-binding (IRP1A_C450S) and one allele that can assume the holoform/aconitase (IRP1A_3R3Q). Finally, we also carried out immunolabelling of the IRP1A_3R3Q transgenic lines, which showed that this protein enters the prothoracic gland nuclei, consistent with the idea that it is assuming the holoform. We chose not to show the latter, because IRP1A_3R3Q overexpression also affected the morphology of the prothoracic glands, which is something we don't understand at the moment.

Furthermore, given the simultaneous introduction of 3 mutations, along with the proposal that the 3R3Q mutant represents holo-IRP1 (with FeS cluster), raises the question as to whether this mutant can efficiently (or at all) incorporate an Fe-S cluster in cells and whether or not it can fold properly. Use of this mutant provides a key basis for several findings but it seems likely that it lacks both RNA binding and aconitase activities and the extent to which it can incorporate an Fe-S cluster is not clear which all together clouds the interpretation of some of the results. In this regard, the inability of the 3R3Q mutant to rescue the loss-of-AGBE-function phenotypes (line 165) and the suggestion that the aconitase function is not sufficient for rescue should be reconsidered.

See our above comments, since they address this concern also.

In addition, in some cases the authors appear to use a 3R3K mutant (line 143 and figure 2B) but in most other cases its 3R3Q. Is this correct? If so, please explain. If not, then then the authors need to demonstrate that the 3R3Q mutant interacts with AGBE by co-IP.

The S2 cell protein-protein interactions experiments were done early in our studies when we were not fully aware of the best possible routes to take. We have now repeated the experiments with 3R3Q for consistency and replaced the relevant figure and text section with the new data. See the new Figure 2A for this, which shows that AGBE interacts with 3R3Q, but not C450S.

Also, on line 143 the authors refer to a A793K mutant. Is 793 not an R?

This has now been replaced with new data (and yes, that was a typo back then).

2. The data in figure 2B shows co-IPs of IRP1 wildtype and mutant IRPs with AGBE. Most notably they show that the C450S mutant which should not be able to bind an Fe-S cluster, fails to interact with AGBE. A blot showing the amount of input for IRP1 variants and also AGBE is needed. Mutants of IRP1 that cannot bind a FeS cluster are unstable and do not accumulate as much as the wildtype IRP1. Is the lack of interaction of the C450S mutant of IRP1 shown in Figure 2B affected by the steady state level of this variant?

We have re-done all experiments regarding co-IPs (see new Figures 2A, 2B, 4E and 4F) and added the requested controls that show input and flow-through. IRP1A_C450S is stable in our hands.

More direct evidence is needed that it is the holo-form of IRP1 that binds to AGBE.

In this regard, are the co-IPs quantitative or is some of the (over?)expressed IRP1 mutants not brought down by the AGBE immunoprecipitation? If this is not quantitative then the extent to which these mutants bind a cluster and whether cluster-containing vs. cluster-devoid species of the mutants is present could affect the results and interpretation.

We now provide extended data regarding protein-protein interactions in S2 cells (see new Figures 2A, 2B). The new figures 4E and 4F show that IRP1A also pulls down Cisd2, but that a triple transfection of AGBE, IRP1A and Cisd2 results in much stronger Cisd2 immunoprecipitation compared to when AGBE is not added. Also see Figure 7C, which shows that overexpression of IRP1A, but not IRP1B results in reduced aconitase activity when overexpression occurs in an AGBE mutant background. While we have not quantified the interactions (other than normalizing the loaded protein amounts), we do provide a negative control (variously tagged eGFP alleles) for all bait proteins, which did not result in

any significant levels of co-immunoprecipitation.

3. The authors provide intriguing data that IRP1 may drive regulation of transcription of genes involved in iron dependent processes. The data in table 2 compares IRP1A vs IRP1A(3R3Q) and notes the much stronger impact of the mutant. Since the 3R3Q mutant used likely lacks both RNA binding and aconitase functions the interpretation of these interesting findings requires reevaluation as does the model in figure 5 which suggests a role for the aconitase function of IRP1 in the nucleus.

As outlined in our data above and in the attached figure, we provide strong evidence that the IRP1A_3R3Q allele is not only functional but also exhibits full or substantial aconitase activity. See also our data on our IRP1A_4R4Q allele (Figure S7), which indeed dramatically reduces aconitase function.

4. The authors provide interesting suggestions about the possible tissue specific functions of IRP1 (i.e. line 215). Some comments are needed about the tissue specific expression of AGBE in flies and other species.

We have not done a consistent AGBE expression analysis yet. However, data-mining (e.g. here <http://flyatlas.org/atlas.cgi?name=CG33138-RA>) suggest that the AGBE mRNA is expressed in most tissues, but that expression levels vary substantially (with testis being the lowest and adipose tissue the highest expression). Our preliminary data on the protein levels are consistent with this wide-spread expression. Anyhow, to reflect this in the text, we have added the following sentence (underlined): “(In this report, we demonstrated that the *Drosophila* glycogen branching enzyme, AGBE, has hitherto undiscovered and essential roles in the regulation of cellular iron homeostasis). We expect that AGBE’s role in iron is not limited to the PG, since genome-wide expression profiling indicates that AGBE is widely expressed³¹.” (lines 321-324).

5. Protein-protein interaction analysis by MS provides important support for the genetic interactions between AGBE, IRP1 and mitoNEET. The authors comment that IRP1 is found to interact with translation factors and also with 40 ribosomal proteins. The latter seems surprising, are the ribosomal protein interactions all direct? If not, or if this is not examined, it should be noted.

Given the sheer number of co-immunoprecipitated proteins, we had to select to most promising candidates for validation. For this study, we picked AGBE and Cisd2 as IRP1A interactors. We are now pursuing the validation of the histones for a future study. As far as the ribosomal proteins go, I would suspect that they are mostly caused by indirect interactions, but they are consistent with the role of IRP1 in translational control. I imagine that the IRP1-bound mRNA is in close vicinity (or even “parked”) at the ribosome, resulting in a high level of pulled down ribosomal components.

The findings that the C450S mutant of IRP1, which cannot be an aconitase, does not interact with AGBE supports the view that the aconitase form is the relevant species. However, without demonstration that it is functional in some way (ie. it can load a FeS cluster), the use of the 3R3Q mutant does not address this important issue.

Please see our above comments about the validity of the 3R3Q allele.

6. Several comments are made in the manuscript about the presumed complete conversion of the aconitase form of IRP1 to the RNA binding form in iron deficiency. The Fe-S cluster of cytosolic aconitase is rather stable and the impact of iron deficiency does not necessarily lead to complete conversion to the RNA binding form

Understood. However, I can't see a reference in our text where we imply that the conversion has to be complete. Please let us know which section (if applicable) you would like to have modified to reflect this.

Reviewer #2 (Remarks to the Author):

Authors have identified an important molecular mechanism that regulates iron concentration and haem-containing proteins in the cell. It is fascinating that a protein known for more than a decade for its function in branching glycogen chains is involved in iron regulation. Although recent findings would give us a hint about this mechanism, no evidence has been provided (articles related to RBCK1 and polyglyglucosan body myopathy). I was surprised authors didn't pick this and mentioned in the discussion.

Thank you for pointing out this link. We indeed were not aware of this! When I looked into this, I got really excited, since one of our genes we work on is RanBP3, for which RNAi also causes porphyria in the prothoracic gland, suggesting parallels exist to RBCK1 (RanBP-type and C3HC4-type zinc finger-containing protein 1). In any case we added the following section to the discussion: "While AGBE has not been directly linked to iron homeostasis, a possible indirect link exists because mutations in *RBCK1* (RanBP-type and C3HC4-type zinc finger-containing protein 1), a gene that encodes an E3 ubiquitin ligase, cause Polyglucosan Body Myopathy, a recently described glycogen storage disorder³². Intriguingly, RBCK1 was shown to control cellular iron homeostasis by degrading the oxidized form of IRP2³³, raising the idea that glycogen and iron processes are linked on multiple levels." (lines 324-329).

GBE has been studied in glycogen metabolism for many years it has been shown that its act together with the activity of glycogen synthase ensures proper branching of glycogen. A slight disturbance in

this delicate balance (towards Glycogen synthase) causes longer glucose chains. Higher activity of glycogen synthase changes the structure of glycogen to starch (Adult polyglucosan body disease APBD, seldom branches) or amylose (Andersen's disease no branches). Polyglucosan (PG) is the apparent product of GBE deficiency. Therefore, it is the hallmark of Glycogenosis Type IV (GSDIV). However, PG also forms in other genetic diseases. Genes that are unrelated to glycogen synthesis or metabolism such as Lafora disease and RBCK1 deficiency. Lafora disease is caused by the deficiency of laforin or Malin (E3 ubiquitin ligase), the other PGB disease is caused by the deficiency of RBCK1 or HOIL1 (Heme-Oxidized IRP2 Ubiquitin Ligase 1). It is still a debate if the PG or energy deficiency is the leading cause of the disease. It is exciting and a great addition to the knowledge in glycogen metabolism and the proteins involved in it, this article brings a third mechanism for the pathogenesis of GSD IV.

Thanks again for this succinct summary!

Authors incorporated all alternative mechanisms and controls. I congratulate for their effort. Besides the addition to the discussion above, I would like to correct a statement made on line 106 “final steps of glycogen synthesis” glycogen synthesis is composed of three steps 1-formation of the primer by glycogenin dimers, 2-extension of glucose chains by glycogen synthase 3- formation of branches by GBE and further extension of branches by glycogen synthase.

It will be better to remove the “final steps.”

We have removed the corresponding text (lines 102-103).

All the protein interactions are shown by immune precipitation and MS. There is no immunostaining and confocal microscopy to back up the data for colocalization of proteins to mitochondria. Proteins may interact with each other in homogenized samples that might be misleading. I suggest authors show at least mitochondrial localization of AGBE. This is not shown before.

While it would be great to have detailed expression data on AGBE, we have instead opted to do an extensive expression analysis of AGBE for a future publication that addresses tissue distribution of AGBE/Cisd2/IRP1A/IRP1B, look how this is affected by dietary iron or iron chelations, subcellular localization, co-localization studies with Cisd2 and IRP1A and IRP1B, and how different mutant background influence this patterns. This is all part of a grant application I submitted in September! (see also next comment).

With respect to AGBE being at the mitochondrion: This is not necessarily a certain prediction, because *Drosophila* Cisd2 lies phylogenetically between mitoNEET (which is found on the outer mitochondrial membrane = OMM) and Naf-1 (which localizes to the OMM, the endoplasmic reticulum (ER) and the

mitochondria-associated membranes). As such, the *Drosophila* Cisd2 may be an amalgamation of both vertebrate proteins, making it hard to predict where the protein will localize at the subcellular level. However, given the link between IRP1A and fly Cisd2, it appears that the ability of mitoNEET to donate Fe-S clusters to IRP1 is conserved in flies (hence the emphasis on mitoNEET rather than Naf-1 in the paper). Our low resolution images of AGBE protein distribution show it to be in the cytoplasm (see picture in reviewer #3 section).

Importantly, we have substantially strengthened the co-IP experiments in the revised version (new Figures 2A, 2B, 4E and 4F), for example, we show that a triple co-transfection of AGBE, IRP1A and Cisd2 is much stronger in immunoprecipitating Cisd2 compared to co-transfecting IRP1A and Cisd2 alone. Taken all data into account, we find that this strongly supports the notion that AGBE and IRP1A physically and genetically interact via Cisd2.

It is also essential to show the function and protein-protein interactions in case of deleterious mutations in AGBE, such as p.Y329S substitution, tyrosine is a conserved amino acid in both vertebrate and Drosophila homologs. This is particularly important because the p.Y329S mutation is in a linker region between amylase and translocase domains, where there is no enzymatic activity is involved. However, it is the most common underlying genetic cause of APBD. Also, there is no information about which part (domain) of the AGBE mediates physical interaction with other proteins.

We were excited to test the Y329S mutation (Y314S in *Drosophila*), but, unfortunately, found that it still interacted with IRP1A. See picture below. We opted to not include the data in the manuscript, because of the negative result. With respect to domains that are critical for IRP1A-AGBE interactions: Also an aim in my grant application! I think it would be exciting to identify potential human alleles of GBE1 that abrogate the interaction with IRP1.

[Redacted]

Reviewer #3 (Remarks to the Author):

This is a very nice and extensive study that begins with a relatively novel screen for knockdowns that cause overgrowth and altered protoporphyrin metabolism in the Drosophila prothoracic gland, and then goes on to make a number of unappreciated and unexpected links between new regulators of iron metabolism. Given the broad general importance of the processes analyzed, and the extensive use of complementary genetic and molecular approaches, it seems highly deserving of publication in Nature communications. There were however some apparent mixups, areas of confusion and omissions that need to be dealt with. I will go through these as they arise in the manuscript.

line 103: I'm confused by "We were unsuccessful in finding independent evidence for the Nos^{IR-X} and spz5IR". Could you clarify/restate what you saw with these knockdowns and how they differed?

We only included the spz5 and Nos^{IR-X} RNAi lines because we wanted to use them as controls for two purposes. For one, neither spz5 and Nos^{IR-X} RNAi animals can be rescued by iron-feeding, emphasizing the fact that AGBE-loss-of-function is unique in this manner (otherwise readers could assume that many or all RNAi lines displaying the porphyria phenotype are rescuable with dietary iron) (see also below). Secondly, we wanted to show that the IRP1A_C450S transgene was functional (because it failed to rescue the AGBE-loss-of function lines). The Nos^{IR-X} line was rescuable to adulthood with IRP1A_C450S, demonstrating that the IRP1A_C450S transgene was fully functional.

To give you a bit more background info, we have now identified ~30 genes where PG-specific RNAi causes the porphyria phenotype in the prothoracic gland. Of these, RNAi against only one of the genes (AGBE) is rescuable with dietary iron. A total of three are rescuable with IRP1A_C450S, and only one (AGBE) can be rescued with the wild type form of IRP1A.

The other RNAi and mutant lines for spz5 and Nos are not showing the porphyria phenotype, and we have spent an extensive amount of energy to validate the specificity of those two RNAi lines. We did not succeed. So, while the porphyria phenotype is real, we cannot rule out off-target effects causing this. Therefore, we only use these lines here to provide context. I hope that clarifies this. Since this is a very complicated matter, I chose not to delve into further detail in the paper itself.

Since the risk of off-targets associated with RNAi, all of our AGBE, Cisd2 and IRP1A/B alleles used in this paper are validated and rely on multiple, independently generated loss-of-function alleles.

Nos, at least, causes the somehow related PG enlargement, but I gather the red colour is due to heme accumulation as opposed to upstream protoporphyrin accumulation or lack of production.

It is, but we can't be sure it is a specific effect that only affects Nos function (see above).

On a related point, if the authors have any idea what the link is between PG size and iron metabolism, if any, it would be helpful for the reader.

We do not really understand this link. At one point, we crossed in a Ras-RNAi construct into several of our lines, since PG-specific expression of constitutively active Ras also results in overgrowth of the ring gland. Ras-RNAi indeed “rescued” the overgrowth phenotype, but at this point we can only speculate as to why the Ras pathway might be hyperactivated in response to a disruption of iron homeostasis (if that is the correct pathway to begin with).

line 104: “however, a second, non-overlapping RNAi line targeting AGBE, AGBEIR2, caused similar phenotypes (Figure 1F).” I assume this means similar to ALAS1 KD - enlarged PG without increased fluorescence (no protoporphyrin)?

Maybe the reviewer accidentally looked at Figure 1E instead of 1F (1E depicts Alas-RNAi)? You can see in 1F that the second RNAi line (AGBE_IR2) also results in protoporphyrin accumulation. We chose to work with the AGBE_IR1 line, because it is overall stronger (4.6% adults vs 58.6%).

The authors should do in situs or RTPCR to confirm that AGBE and IRPI are expressed and when in the PG (unless already published somewhere). So far, they have only looked at Gal4/UAS induced tagged versions of IRP1A/B. Do these levels change in AGBE or other mutant backgrounds? with Fe depletion/supplementation?

As I have outlined in my response to reviewer #2, we have preliminary expression data confirming expression of AGBE in the prothoracic gland (on the protein level it is easily visible in the PG cytoplasm, see picture below), but we are planning an extensive analysis for a future study. Also, we published ring gland-specific microarray data (Ou et al 2016, Cell Reports) and have done many ring gland-specific RNA-Seq experiments since then. They all show strong (mRNA) expression for AGBE, IRP1A and Cisd2.

[Redacted]

I would like to add one detail to address an apparent misunderstanding by the reviewer: We do provide expression data for IRP1A and IRP1B in the manuscript, since we show expression for two knock-in alleles IRP1A_3F and IRP1B_3F, which are not Gal4-induced transgenes (please see Figure 3F and Figure 4J), but rather leave the endogenous regulatory regions completely intact (See Figure S4 for a map of the knock-ins).

line 128: “However, ubiquitous expression of RNAi caused widespread larval lethality, confirming that all RNAi lines were functional” Presumably this is referring to AGBE RNAi lines.

No, we meant all RNAi lines shown in Figure S5. To improve clarity, the sentence now reads: “However, ubiquitous expression of RNAi targeting these glycogen biosynthesis genes caused widespread larval lethality, confirming that all RNAi lines were functional, and that disruption of glycogen biosynthesis per se in the PG did not cause any iron- or haem-related phenotypes, but was a unique feature of AGBE.” (lines 123-126).

134: “A search of protein-protein interaction databases¹⁸ revealed that human GBE1 physically interacts with IRP1” How was this original interaction determined? Was the interaction published and discussed previously, or just annotated?

You can find the result here (<https://thebiogrid.org/108902/summary/homo-sapiens/gbe1.html>) (on BioGrid), Aco1 (aka IRP1) is one of the 29 reported interactors of human GBE1. According to the “details” link, the interaction is based on co-fractionation of macro protein complexes published in Wan, C. et al. (Panorama of ancient metazoan macromolecular complexes. Nature 525, 339-344 (2015). My original reference only referred to the BioGrid paper, and I have now added Wan et al to the sentence (line 130).

The following two quotes from the manuscript are contradictory.

138: Only IRP1A has been shown to switch from holo- to the IRE-binding apoform, while IRP1B is believed to act only as an aconitase .

*148: “This was paradoxical, as AGBE mutations caused iron-deficiency phenotypes, but holo-IRP1A has no known roles in iron homeostasis as it is believed to only act as an aconitase.”
The finding is not paradoxical if line 138 is correct.*

Hmm, I am not quite sure what the reviewer means here. In line 138, we state that IRP1A can switch from apo- to holo-form, and that the apo-form is IRE-binding. We explain elsewhere that it is the IRE-binding (= mRNA-binding form) that is known to regulate cellular iron homeostasis. In line 148, we state that the IRP1A holo-form is only known to act as an aconitase, which was hitherto not linked to a role in iron homeostasis. Therefore, the finding that AGBE interacts with the aconitase form (aka holo-IRP1A) is indeed paradoxical, since AGBE-loss-of function causes iron phenotypes. I left the sentence as is. Our original expectation was that AGBE would interact with the apo-form, not the other way around.

In figure 1, why wasn't IRP1B tested for interaction with AGBE? Some redundancy here could explain the weaker IRP1A phenotype as compared to AGBE (discussed further below).

In Figure 3C we show that IRP1B null mutants are fully viable. Also, IRP1B is believed to act only as an aconitase. We therefore focussed on IRP1A. The reviewer is correct that there is some redundancy, for this please take a look at the attached figure at the end of this document (to be used for a future manuscript). Our mass spec data was reciprocal, by the way, and a tagged IRP1B knock-in allele co-immunoprecipitated AGBE, while a tagged knock-in allele of AGBE co-immunoprecipitated IRP1B. We felt this was solid enough data for a protein that has minor functions.

157: "we observed dramatic rescue of PG>NOSIR-X RNAi animals with respect to both the lethality (Figure 2D) and protoporphyrin accumulation (not shown).

Whats the presumed mechanism/explanation for this? Where does NOS fit in? Protoporphyrin or heme accumulation?

As outlined above in a response to a related question from this reviewer, we used the Nos^{IR-X}-RNAi line solely for contextual information. Since we have not succeeded in validating the specificity of this line [it does knock down Nos, that is published - (Caceres, L. et al. Nitric oxide coordinates metabolism, growth, and development via the nuclear receptor E75. Genes Dev 25, 1476-1485, 2011), but there may be an off-target for this particular RNAi line], I have no intention in pursuing this gene any further. However, Nos is a great fit, because a number of papers have shown that Nitric Oxide (NO) destabilizes the IRP1 Fe-S cluster, and so one of our pet theories is that a pulse of NO (for which we have some evidence that they occur in the PG) could act as a signal to force holo-IRP1A into the apo-form. In fact, this may be the a key mechanism by which a cell can activate IRP1 despite an abundance of iron, since some cells - like PG cells - have vastly higher iron demands than other cells. But until I have independent evidence that another Nos allele recapitulates the porphyria phenotype, this project will remain on ice.

line 175: Since both lines are available, I would like to see double IRP1A/B RNAi - I suspect there's some redundancy happening, and that this would result in a stronger phenotype.

We have done this of course, but since the IRP1A-RNAi is so weak to begin with, the double RNAi didn't change the outcome. But, be sure to take a look of our genetic analysis (figure at the end of this letter), which clearly shows that there is some redundancy. In particular, IRP1A nulls can survive to adulthood on iron-rich food, but taking out one copy of IRP1B abolishes viability (last row of the figure).

line 192: “Taken together, these results showed that only IRP1A has critical roles in iron homeostasis and that IRP1A depletion phenocopied the iron-dependent porphyria seen in AGBE mutants.”

If I understand correctly, I disagree. I don't see that a contribution by IRP1B, perhaps lesser and largely redundant in manner, has been ruled out.

I changed the sentence as follows: “Taken together, these results showed that IRP1A is the principal regulator of cellular iron homeostasis in *Drosophila* and that IRP1A depletion phenocopied the iron-dependent porphyria seen in AGBE mutants.” I think the wording is justified, since IRP1B null mutants are viable with no obvious defects when reared on iron-depleted medium, while IRP1A nulls die under the same conditions.

IRP1B did not require AGBE for nuclear localization, but this does not mean it cannot act like IRP1A in terms of iron regulation. In fact, mutual pull-downs are consistent with all working together with IRP1A/B overlapping functions. If IRP1B can interact with AGBE and IRP1A, it doesn't need to interact with Cisd2 (MitoNEET) directly in order to provide function. Do IRP1s hetero/homodimerize? If not directly, it sounds like they do indirectly via AGBE or MitoNEET.

We now show that AGBE directly interacts with Cisd2, and that a co-transfection of IRP1A, Cisd2 and AGBE results in synergistic interaction (new Figure 4E, F). The other suggestions are entirely possible, and will be looked at in future experiments!

There appear to be significant issues with the MS data. The s1 and s2 tables referred to for MS data do not appear to be included.

Perhaps the reviewer had an issue with accessing the files? Everything on my end seems to indicate that the submission was complete. I just checked again, and Tables S1 and S2 are online. However, and this may be the reason, they were not posted together with the Tables S3-S6, because the latter are all Excel files. Tables S1 and S2 are simple text tables, and were part of the article PDF file that comprises the manuscript, all main figures, all supplemental figures and supplemental Tables S1 and S2 (which can be found on pages 44 and 45 of the original submission).

Tables S3 and S4 don't clearly indicate the bait used and where it shows in the recovered peptides/spectra.

In our defense, that is why we provided Table S2, to provide an exact guide as to which protein/allele/strain was used and what the source tissue was (basically either PG cells or whole larvae). However, the second row of Table S3 does indicate the bait protein, except for the controls where this does not apply (the control samples do not express the bait protein).

Proteins generally weren't named, so I could not tell if the proteins mentioned in the results were there in the tables.

Agreed! While the protein names do appear in the “description” column, it is not great for quickly scanning the data. Therefore, I have added two columns for each of the two worksheets in Table S3: one for the full gene name and one for the gene symbol.

Legends to explain the tables appear to be missing.

There are/were three worksheets in the excel file for table S3. The last contains the legend. Also, as outlined above, Table S2 provides a comprehensive summary for Table S3.

The most abundant proteins look like chaperones and other abundant proteins such as ribosomal subunits and histones (actually didn't see the latter).

That's strange, because the histones were labelled all in green (in Table S3, both worksheets). Green highlights refer to the proteins shown in Figure 4A (for legend, see the last tab for worksheet #3).

To be real, interactors should be less abundant than the bait, which if well done, should be the most abundant protein in the purification. Anything more abundant is almost certainly background.

As a general rule, yes, but not always necessarily. That entirely depends on the stoichiometry. A single bait molecule could interact with two molecules of the same time, in which case one would pull down up to twice the number of the bait protein molecules. We discussed this with our experts in the MS core facility, and they see this all the time.

Comments on data analysis methods (spectral counts, non-specific protein filtering etc) should also be mentioned briefly in the results section.

I have added the following sentence to this effect: “As controls, we used a total of five wild type samples (which lack Flag-tagged proteins), processed them in parallel to the experimental samples, and removed all proteins found in the controls from the experimental data sets (Table S3).”

We have also expanded our Methods section and added the following:

“In brief, we performed overnight in-gel trypsin digestion following standard procedures. Fractions containing tryptic peptides were resolved and ionized using nanoflow HPLC coupled to an LTQ XL-Orbitrap hybrid mass spectrometer. The mass spectrometer was operated in data-dependent acquisition mode, recording high-accuracy and high resolution survey Orbitrap spectra using external

mass calibration. The fourteen most intense multiply charged ions were recorded in the linear ion trap, which gave an ion score. The sum of score from peptides of the same protein represents score for that protein. Data was analyzed using Proteome Discoverer 1.4 and a Uniprot *Drosophila* proteome database was searched using SEQUEST software.”

For future experiments, the authors should consider the possibility that the ability of IRPIs to bind IREs could facilitate interactions with guide RNAs that play a role in nuclear localization and target gene promoter selection. It would be interesting to do RNAseq on affinity purified IRIPs (RIPseq). If RNAs are found, check for IRE's. It should also be considered that many of the interactions detected may be indirect via linking RNA and additional proteins. I think that doing pull-downs in the presence of RNase, and trying direct interactions with purified proteins should be considered (or minimally, the above possibility acknowledged).

That is also an aim in my current grant proposal! Thanks!

Reviewer #4 (Remarks to the Author):

The manuscript by Huynh et al. uncovers two novel findings, using Drosophila as model organism: (1) AGBE plays a role in maintaining functional IRP1A, likely involving also Cisd2, and (2) IRP proteins translocate to the nucleus, likely in their holo-forms, where they interact with various histone proteins. Finding (1) reveals a previously unknown link between iron homeostasis and glucose metabolism, and finding (2) leads to the suggestion that IRPs in the nucleus might play, via their aconitase activity, a role in regulating the supply of acetyl-CoA for histone acetylation.

The study encompasses a range of transgenic and other modified fly lines, immunostaining, transcriptomic data, and the determination of interactomes for several proteins, for both whole-body and the specific tissue under study (prothoracic gland).

Together, the data support the major findings of the study. The authors have also considered alternative causal relationships, e.g. the possibility that the observed interactions between AGBE and IRPs relates to their (cytosolic) aconitase function.

Although this might be outside the scope of the study, it would have been desirable to have more biochemical information on how the interactions with AGBE might take place, and what the “moonlighting” function of AGBE might be.

Agreed! We have expanded and improved the protein interaction studies (New Figures 2A-b and 4E-F), but beyond that we feel that there are so many avenues to pursue that we need to stay focused. As

outlined in a response to the other reviewers, we are already planning to map the domains responsible for the IRP1A-AGBE interaction, and the AGBE-Cisd2 interaction as well. Plus, we need to catalog when and where these major players are active, and how they respond to changes in dietary iron etc.

In summary, the findings are novel and are likely to influence and stimulate work in this field. A few suggestions for clarifications/improvements are given below.

The interactomes of IRP1A, IRB1B and AGBE were determined by co-immunoprecipitation followed by SDS-PAGE and MALDI-TOF MS. The authors studied a large number of carefully selected samples to address various detailed questions. Interactome data are summarised in Figure 4, and evaluated MS data are presented in a supplementary data file.

The Excel data file contains scores that refer to abundances, but it is unclear how these were determined. It would therefore be desirable to provide more detail on sample preparation for MS, and especially the quantitative aspect in these data, on page 21. This should include information on detection limits.

Upon completion of trypsin digestion, peptides were resolved and ionized using nanoflow HPLC coupled to an LTQ XL-Orbitrap hybrid mass spectrometer. We looked for the most intense multiply charged ions, which were then sequentially fragmented by using collision-induced dissociation, and spectra of their fragments were recorded in the linear ion trap which give us an ion score. Scores of all detected peptides from the same protein were added up together and represented the score of that protein. We have provided more information in the methods section.

It may also be noted that all “micro” signs seem to have been converted to another symbol.

Fixed, and hopefully doesn't re-occur upon submission.

Line 143: There seems to be an “A” missing in IRP1^{A793K}.

That line is gone now - replaced with new experiments.

[Redacted]

Reviewers' comments:

Reviewer #1 (Remarks to the Author):

The revised manuscript by Huynh focuses on the unanticipated coordinative role of IRP1, the glycogen branching enzyme AGBE and a member of the mitoNEET family in iron homeostasis in *Drosophila*. Through the use of a range of genetic approaches along with biochemical and cell biological methods the authors provide clear evidence that AGBE has a wholly unexpected role in iron metabolism via its interaction with IRP1, a cytosolic iron regulated protein controlling mRNA fate. The work herein provides support for novel actions of IRP1 in the nucleus. Overall, the implications of the work remain potentially far reaching and paradigm shifting.

1. The authors provide evidence that the 3R3Q mutant of IRP1 has aconitase activity which strengthens the manuscript. However, it remains unclear if this mutant is an aconitase only form. New questions also arise given the surprising finding that this mutant has aconitase activity.

a. The 3 arginines mutated here are among the highly conserved ~20 residues found in the active site of aconitases including those in *E. coli*. The R residues mutated here for the 3R3Q mutant have all been proposed by crystallographic and (some) mutational analyses (Lauble et al, *Biochem.* 31:2735; Lauble and Stout, *Proteins* 22:1 and Zheng et al *JBC* 267:7895) to be involved in either substrate recognition or other aspects of enzyme function. On this basis it is surprising that the 3R3Q mutant retains essentially 100% of the aconitase activity for *Drosophila* 1A and 1B isoforms. Some issues arise from this finding.

First, if these highly conserved residues previously identified mutationally or structurally as being involved in the enzymatic activity of aconitase do not have the same role in the *Drosophila* enzyme then by extension one wonders if they are required for RNA binding as shown by Philpott using individual mutations. Given these observations and the centrality of the 3R3Q mutant in the current manuscript it seems key that it be determined if this mutant of aconitase 1A can still bind RNA with an affinity similar to the wildtype protein. The authors refer to 3R3Q as "aconitase only" (line 159) and also as "presumed to have" RNA binding activity (line 302)

Second, there are some questions about the aconitase activity shown in figure S7. The data is not sufficiently quantitative to allow the conclusion (eg. Line 141) that 3R3Q retains full aconitase activity.

What is relative activity (the y-axis label)?

Are these values normalized to the level of expression of the transgene protein. If not, they need to be.

Given that at least one of the R residues is involved in substrate recognition, and it's not clear what the substrate concentration was used for the *in vitro* assays, this needs to be addressed. Furthermore, can the authors exclude the possibility that the 3R3Q mutant is not an active aconitase at *in vivo* substrate concentrations since it's likely the enzyme assay from OxisResearch used saturating concentrations of substrate. If the K_m is substantially increased in this mutant it may not have relevant activity *in vivo*.

2. The authors use CoIP followed by Westerns or MS to identify proteins. New controls were added as requested which helps. Some questions still arise.

It's not clear from the figure 2A/B or the legend what % of the input is shown in the "Input lane".

It's not completely clear that AGBE binds the aconitase form.

The CoIPs show that AGBE antibodies bring down AGBE and various versions of IRP1A. It appears that some fraction of IRP1A is brought down as the co-IP lane is lighter than then the input for 1A (and higher for AGBE). Without knowledge of what fraction of the overexpressed IRP1A proteins is in the aconitase (holo) form its not clear whether one can conclude that AGBE binds the aconitase (holo) form especially since it appears that a significant fraction of IRP1A is not brought down. The C450S results argues its not the RNA binding form but its not clear that all of overexpressed 3R3Q is in either holo vs apo as some could be improperly folded. If you IP AGBE can you bring down aconitase activity?

A blot is needed to demonstrate the level of expression of the transgenes relative to the endogenous proteins.

3. Regarding the aconitase activity – see above.

4. Okay.

5. See above re: aconitase issues. The new data does show 3R3Q can have aconitase activity in vitro but does not address whether the mutations affect substrate affinity and in vivo relevance nor, as noted in 2, what fraction of this mutant when expressed loads cluster.

6. It's the general tone of the manuscript (and the field). Also, one can induce significant RNA binding by IRP1 in the face of no detectable change in aconitase activity in response to iron deficiency since in some systems the aconitase form is in great excess to the RNA binding form. For instance – line 246

Some new points

7. It would help in the manuscript if the first time you mention 3R3Q you explicitly list the residue numbers.

8. The authors refer to nuclear import of holo-IRPs such as on line 204. At this point its really nuclear accumulation.

9. Line 372. "Imported" implies transporter. Is there evidence that the citrate transporter is in the nuclear membrane? Also, it could also just diffuse through the nuclear pores.

10. Line 407. Not everyone would agree that IRP knockouts have subtle phenotypes in normal conditions. Some report (two labs) clear neurological impacts of IRP2 deficiency while others (two labs also) find substantial dysregulation of erythropoiesis in IRP1 deficient mice fed a normal diet.

11. Line 145 – I think you mean 1B not 1A.

12. "is essential." Should be "is essential in Drosophila." Line 286

Reviewer #2 (Remarks to the Author):

Authors have answered my concerns and did the experiments I suggested. I congratulate their efforts and the information they add to the pathology of GSD IV.

Reviewer #4 (Remarks to the Author):

In my opinion, and as far as my comments are concerned, the authors have addressed these comprehensively. I also note that a substantial amount of new data have been included that have provided more clarity and better support for the major conclusions of the study.

Reviewer #5 (Remarks to the Author):

Some minor modifications are suggested

1. original comment

line 103: I'm confused by "We were unsuccessful in finding independent evidence for the NosIR-X and spz5IR". Could you clarify/restate what you saw with these knockdowns and how they differed?

The authors addressed this question, but to help the reader, please elaborate a bit on this in the text regarding the possibility of off-targeting and the possible reason for IRPA rescue.

2. original comment

line 128: "However, ubiquitous expression of RNAi caused widespread larval lethality, confirming that all RNAi lines were functional"....

RNAi causing lethality does not necessarily mean it is truly against the intended target. RNAi off-targeting can also cause lethality. The original sentence read as if all RNAi lines are real and disrupt glycogen biosynthesis. Please restate the sentence.

3. original comment

Line 134: "A search of protein-protein interaction databases¹⁸ revealed that human GBE1 physically interacts with IRP1" How was this original interaction determined? Was the interaction published and discussed previously, or just annotated?

Please provide the link or the web site in the text.

Response to reviewers.

Reviewers' comments:

Reviewer #1 (Remarks to the Author):

The revised manuscript by Huynh focuses on the unanticipated coordinative role of IRP1, the glycogen branching enzyme AGBE and a member of the mitoNEET family in iron homeostasis in Drosophila. Through the use of a range of genetic approaches along with biochemical and cell biological methods the authors provide clear evidence that AGBE has a wholly unexpected role in iron metabolism via its interaction with IRP1, a cytosolic iron regulated protein controlling mRNA fate. The work herein provides support for novel actions of IRP1 in the nucleus. Overall, the implications of the work remain potentially far reaching and paradigm shifting.

1. *The authors provide evidence that the 3R3Q mutant of IRP1 has aconitase activity which strengthens the manuscript. However, it remains unclear if this mutant is an aconitase only form. New questions also arise given the surprising finding that this mutant has aconitase activity.*

a. *The 3 arginines mutated here are among the highly conserved ~20 residues found in the active site of aconitases including those in E. coli. The R residues mutated here for the 3R3Q mutant have all been proposed by crystallographic and (some) mutational analyses (Lauble et al, Biochem. 31:2735; Lauble and Stout, Proteins 22:1 and Zheng et al JBC 267:7895) to be involved in either substrate recognition or other aspects of enzyme function. On this basis it is surprising that the 3R3Q mutant retains essentially 100% of the aconitase activity for Drosophila 1A and 1B isoforms. Some issues arise from this finding.*

First, if these highly conserved residues previously identified mutationally or structurally as being involved in the enzymatic activity of aconitase do not have the same role in the Drosophila enzyme then by extension one wonders if they are required for RNA binding as shown by Philpott using individual mutations. Given these observations and the centrality of the 3R3Q mutant in the current manuscript it seems key that it be determined if this mutant of aconitase 1A can still bind RNA with an affinity similar to the wildtype protein. The authors refer to 3R3Q as “aconitase only” (line 159) and also as “presumed to have” RNA binding activity (line 302)

I disagree that one can conclude that *Drosophila* IRP1A is somehow different from other IRP1 aconitases. The reason is that the aconitase function of eukaryotic IRP1 has never been sufficiently addressed by mutational analysis. Therefore we are the first to test the aconitase function of IRP1 by mutational analysis of these specific arginines. The data provided in Zheng et al (1992) are based on mitochondrial aconitase, not IRP1. Likewise, structural studies identified amino acid residues that participate (under the conditions in the experiment) in substrate recognition etc., but they do not demonstrate whether those residues are functionally required. The aconitase assays conducted by Philpott et al are limited to the C437S, C503S and C505S mutations. So, as far as I can tell, no one has measured the effect of the R-Q mutations on

the IRP1 aconitase activity, and existing papers refer to the “active site” that has been defined by crystallographic and mutational studies of mitochondrial (and *E. coli*) aconitase. Intriguingly though, one of the key residues required for mitochondrial aconitase activity, Arg580 (Zheng et al, 1992), is not mutated in our IRP1A^{3R3Q} mutant (which has Aconitase activity), but this residue is mutated in our IRP1A^{4R4Q} (i.e., all three mutations present in IRP1A^{3R3Q} plus the equivalent Arg580 mutation), and this abolishes aconitase activity (Fig. S7). This aspect is in strong agreement with the Zheng et al paper.

In any case, since the published mutational and structural data were of limited use here, we addressed this directly by experimental means. For this, we used our IRP1A^{3R3Q} and IRP1A^{C450S} CRISPR knock-in alleles (now added to the manuscript, modified Fig. S4), and conducted quantitative RNA-immunoprecipitation (RIP) and aconitase assays (new Figs. 2C, D). We normalized samples relative to the tagged protein levels. This was done by loading immunoprecipitated IRP1A variants on an SDS-PAGE followed by Western blotting (see detailed addition to our method section, highlighted), and analysing band intensity via ImageJ. We then amplified one of the co-immunoprecipitated mRNAs, a confirmed IRE-containing transcript of the *SdhB* gene, via real-time PCR. This confirmed that IRP1A^{C450S} has 4-fold higher RNA-binding than corresponding extracts with wild type IRP1A and >150-fold RNA-binding than the IRP1A^{3R3Q} form (new Fig. 2C). Further, extracts from animals carrying the knock-in IRP1A^{3R3Q} allele had ~70% of the aconitase activity of extracts from wild type larvae (new Fig. 2D). This corroborates our data presented in the previous revision, where we showed that transgenically produced IRP1A^{3R3Q} had significant aconitase activity (Fig. S7). Also, we showed that transgenically produced wild type IRP1A has significantly reduced aconitase activity when expressed in an *AGBE*^{-/-} mutant background, consistent with our finding that this causes cytoplasmic accumulation of IRP1A.

Second, there are some questions about the aconitase activity shown in figure S7. The data is not sufficiently quantitative to allow the conclusion (eg. Line 141) that 3R3Q retains full aconitase activity.

What is relative activity (the y-axis label)?

Are these values normalized to the level of expression of the transgene protein. If not, they need to be.

Given that at least one of the R residues is involved in substrate recognition, and it's not clear what the substrate concentration was used for the in vitro assays, this needs to be addressed. Furthermore, can the authors exclude the possibility that the 3R3Q mutant is not an active aconitase at in vivo substrate concentrations since it's likely the enzyme assay from OxisResearch used saturating concentrations of substrate. If the Km is substantially increased in this mutant it may not have relevant activity in vivo.

First off, the OxisResearch kit allows for a maximum of 25 nmol/min activity, and the highest we measured in all of our experiments was the IRP1A^{WT} activity in our S2 cell assay (Fig. S7A), which reached 17 nmol/min. As such, we do not believe that the transgenic levels of our IRP1A variants have saturated the OxisResearch kit.

Secondly, as outlined above, we have now measured the aconitase activity produced by the knock-in alleles IRP1A^{C450S} and IRP1A^{3R3Q} (Figure 2D). Since these alleles behave like the endogenous counterparts (all regulatory sequences are present and intact), no overexpression issues arise that may artificially saturate the aconitase assay (even though we did not have this issue in the first place). To remove background aconitase activity, we crossed the *IRP1A* alleles into an *IRP1B*^{-/-} mutant background, and the mitochondrial contribution was removed (or significantly reduced) via ultracentrifugation. The activity was then compared to the control samples (= 100%). Both the measurements in Fig. S7 and Fig. 2D are normalized to the transgenic or knock-in-derived protein. In case of Fig. S7, we evaluated the levels of transgenic protein by immunoprecipitating IRP1 and scanning the resulting Coomassie bands from SDS-PAGE followed by ImageJ analysis. For Fig. 2D, we used a similar approach but labelled the immunoprecipitated IRP1A proteins by Western blotting, scanning, and ImageJ quantification.

2. *The authors use CoIP followed by Westerns or MS to identify proteins. New controls were added as requested which helps. Some questions still arise.*

Its not clear from the figure 2A/B or the legend what % of the input is shown in the "Input lane".

In all cases, the input lane represents 10% of the lysate is used for the input lane. Now added to the figure legend (highlighted).

Its not completely clear that AGBE binds the aconitase form.

The CoIPs show that AGBE antibodies bring down AGBE and various versions of IRP1A. It appears that some fraction of IRP1A is brought down as the co-IP lane is lighter than then the input for 1A (and higher for AGBE). Without knowledge of what fraction of the overexpressed IRP1A proteins is in the aconitase (holo) form its not clear whether one can conclude that AGBE binds the aconitase (holo) form especially since it appears that a significant fraction of IRP1A is not brought down. The C450S results argues its not the RNA binding form but its not clear that all of overexpressed 3R3Q is in either holo vs apo as some could be improperly folded. If you IP AGBE can you bring down aconitase activity?

I agree that it is not 100% certain that it is the aconitase form that interacts with AGBE, and that it is possible that a third, non-RNA and non-Aconitase form may interact with AGBE. However, I do not find it surprising that the interaction between the two proteins, in these experimental conditions, is not 100%, and that only a fraction of the IRP1A binds to AGBE. In fact, this is entirely normal for this kind of assay and in fact expected. This could be for any number of reasons (folding of IRP1A or AGBE, affinity for each other, or IRP1A assuming the apo-form

and holo-form under these conditions). So, in other words, the data is entirely consistent with AGBE binding to the aconitase-only form. I think the key observation is that AGBE does not interact with the apo-form (IRP1A^{C450S}), demonstrating that it interacts with the holo-form or, much less likely, with an unknown third form. In case of the latter, however, one would have to postulate that the C450S mutation not only abolishes the holo-form, but also the hypothetical third conformation. Taken together, this seems extremely unlikely to me. With respect to the IRP1A^{3R3Q} form, it does have aconitase function and little or no RNA-binding activity, and it follows that it can assume the holo-form. Consistent with this observation, we observe physical interaction between IRP1A^{3R3Q} and AGBE.

In the manuscript, we say that AGBE binds to the holo-form, and we do not mean this to be synonymous with the term “aconitase”. This is based on the fact that the C450 amino acid is essential for a) binding the Fe-S cluster and b) interaction with AGBE. Whether the holo-form has different sub-classes remains to be seen.

Unfortunately, it is not technically feasible to immunoprecipitate AGBE to co-IP sufficient amounts of IRP1A to conduct aconitase assays. Plus, it could always be argued that the co-immunoprecipitated IRP1A assumes aconitase activity in the reaction tube, but not *in vivo*.

A blot is needed to demonstrate the level of expression of the transgenes relative to the endogenous proteins.

3. *Regarding the aconitase activity – see above.*
4. *Okay.*
5. *See above re: aconitase issues. The new data does show 3R3Q can have aconitase activity in vitro but does not address whether the mutations affect substrate affinity and in vivo relevance nor, as noted in 2, what fraction of this mutant when expressed loads cluster.*

See above.

6. *It's the general tone of the manuscript (and the field). Also, one can induce significant RNA binding by IRP1 in the face of no detectable change in aconitase activity in response to iron deficiency since in some systems the aconitase form is in great excess to the RNA binding form. For instance – line 246*

Again, I agree, but I don't think we ever emphasize complete conversion anywhere in the manuscript. After all, many biological processes do not operate in an all-or-nothing fashion, so I don't feel particularly compelled to make any changes. I left the text as is.

Some new points

7. *It would help in the manuscript if the first time you mention 3R3Q you explicitly list the residue numbers.*

Done. Lines 152-153. Also, since the manuscript now features the IRP1A^{C450S} and IRP1A^{3R3Q} knock-in alleles, we have added the alleles to Figure S4, allowing the reader to see the relative location of these residues.

8. *The authors refer to nuclear import of holo-IRPs such as on line 204. At this point its really nuclear accumulation.*

Good point. I changed “import” to “accumulation” (now line 225).

9. *Line 372. “Imported” implies transporter. Is there evidence that the citrate transporter is in the nuclear membrane? Also, it could also just diffuse through the nuclear pores.*

Not sure, but it could definitely be a passive process. I changed the wording as follows: “Nuclear citrate is converted to acetyl-CoA and oxaloacetate by nuclear ATP-citrate lyase (ACL)...”

10. *Line 407. Not everyone would agree that IRP knockouts have subtle phenotypes in normal conditions. Some report (two labs) clear neurological impacts of IRP2 deficiency while others (two labs also) find substantial dysregulation of erythropoiesis in IRP1 deficient mice fed a normal diet.*

The sentence now reads: “similar to *Drosophila* IRP1A, null mutations of mouse IRP1 or IRP2 are non-lethal under normal conditions, however, the double knockout is embryonic lethal”

11. *Line 145 – I think you mean 1B not 1A.*

No. The emphasis here is on “holo-IRP1A” not “IRP1A”. As I mentioned above, I think it is safer for this study to distinguish between “holo-IRP1” and “aconitase”, since future studies will have to address whether nuclear holo-IRP1 has novel functions. So, I do not think that the two terms should be considered synonymous (for now). But then again, it is possible that the transcriptional role of holo-IRP1 is entirely effectuated through its aconitase function, but future studies will have to address this.

12. *“is essential.” Should be “is essential in Drosophila.” Line 286*

Done (now line 292)

Thank you for all the help!

Reviewer #2 (Remarks to the Author):

Authors have answered my concerns and did the experiments I suggested. I congratulate their efforts and the information they add to the pathology of GSD IV.

Thanks!!

Reviewer #4 (Remarks to the Author):

In my opinion, and as far as my comments are concerned, the authors have addressed these comprehensively. I also note that a substantial amount of new data have been included that have provided more clarity and better support for the major conclusions of the study.

Much appreciated - thank you.

Reviewer #5 (Remarks to the Author):

Some minor modifications are suggested

1. original comment

line 103: I'm confused by "We were unsuccessful in finding independent evidence for the Nos^{IR-X} and spz5^{IR}". Could you clarify/restate what you saw with these knockdowns and how they differed?

The authors addressed this question, but to help the reader, please elaborate a bit on this in the text regarding the possibility of off-targeting and the possible reason for IRPA rescue.

The sentence now reads: We were unsuccessful in finding independent evidence for the Nos^{IR-X} and spz5^{IR} lines, suggesting that the phenotypes were caused by off-target effects.

2. original comment

line 128: "However, ubiquitous expression of RNAi caused widespread larval lethality, confirming that all RNAi lines were functional"....

RNAi causing lethality does not necessarily mean it is truly against the intended target. RNAi off-targeting can also cause lethality. The original sentence read as if all RNAi lines are real and disrupt glycogen biosynthesis. Please restate the sentence.

Agreed. The sentence now reads: "However, ubiquitous expression of RNAi targeting these glycogen biosynthesis genes caused widespread larval lethality, confirming that all RNAi transgenes in these lines were expressed, which suggested that disruption of glycogen

biosynthesis per se in the PG did not cause any iron- or haem-related phenotypes, but was a unique feature of *AGBE*.”

I would like to add that not only did we test all other glycogen synthesis genes, but also all available RNAi lines - we never saw autofluorescing ring glands, except for *AGBE*.

3. original comment

*Line 134: “A search of protein-protein interaction databases¹⁸ revealed that human GBE1 physically interacts with IRP1” How was this original interaction determined? Was the interaction published and discussed previously, or just annotated?
Please provide the link or the web site in the text.*

In the revision, we added citation #19 (Wan, C. et al. Panorama of ancient metazoan macromolecular complexes. *Nature* 525, 339-344 (2015)), which was the study that reported this interaction. As you can see from the title this interaction came out of a big proteomic study and was not individually featured in any study prior to ours. I added a footnote to the text containing the hyperlink to the BioGrid database.

Reviewers' comments:

Reviewer #1 (Remarks to the Author):

The authors have responded to the previous inquiries and provided additional data that further strengthens the manuscript

The work described will have a significant impact on the field and is clearly paradigm shifting. The proposed work will have far-reaching impacts on other in the iron field as well as those interested in glycogen storage diseases, moonlighting roles of enzymes and has the potential to re-orient thinking regarding the clinical overlaps between diseases of iron and carbohydrate metabolism

Reviewer #5 (Remarks to the Author):

The authors did what I suggested. I have no further comments.